# The potential of inversions to accumulate balanced sexual antagonism is supported by simulations and *Drosophila* experiments

Christopher S McAllester*, John E Pool*

Laboratory of Genetics, University of Wisconsin, Madison, United States

## eLife Assessment

This study proposes a new model that could solve some long-standing puzzles about inversion polymorphisms in *Drosophila melanogaster* by invoking sexually antagonism and negative frequency-dependent selection. While the idea developed here is a **valuable** contribution to the field, the experiment only addresses one element of the hypothesis, so that the empirical evidence in support of the model remains **incomplete**.

**\*For correspondence:**
cmcallester@gmail.com (CSMcA);
jpool@wisc.edu (JEP)

**Competing interest:** The authors declare that no competing interests exist.

**Abstract** Chromosomal inversion polymorphisms can be common, but the causes of their persistence are often unclear. We propose a model for the maintenance of inversion polymorphism, which requires that some variants contribute antagonistically to two phenotypes, one of which has negative frequency-dependent fitness. These conditions yield a form of frequency-dependent disruptive selection, favoring two predominant haplotypes segregating alleles that favor opposing antagonistic phenotypes. An inversion associated with one haplotype can reduce the fitness load incurred by generating recombinant offspring, reinforcing its linkage to the haplotype and enabling both haplotypes to accumulate more antagonistic variants than expected otherwise. We develop and apply a forward simulator to examine these dynamics under a tradeoff between survival and male display. These simulations indeed generate inversion-associated haplotypes with opposing sex-specific fitness effects. Antagonism strengthens with time, and can ultimately yield karyotypes at surprisingly predictable frequencies, with striking genotype frequency differences between sexes and between developmental stages. To test whether this model may contribute to well-studied yet enigmatic inversion polymorphisms in *Drosophila melanogaster*, we track inversion frequencies in laboratory crosses to test whether they influence male reproductive success or survival. We find that two of the four tested inversions show significant evidence for the tradeoff examined, with *In(3 R)K* favoring survival and *In(3 L)Ok* favoring male reproduction. In line with the apparent sex-specific fitness effects implied for both of those inversions, *In(3 L)Ok* was also found to be less costly to the viability and/or longevity of males than females, whereas *In(3 R)K* was more beneficial to female survival. Based on this work, we expect that balancing selection on antagonistically pleiotropic traits may provide a significant and underappreciated contribution to the maintenance of natural inversion polymorphism.

## Introduction

### The enigmatic prevalence of inversion polymorphism

Inversions, DNA segments in reversed sequence order relative to the ancestor, have been a focus of population genetic study for their direct or indirect effects on recombination in natural populations for more than a century. In their most dramatic effect on meiosis, inversions generate aneuploid gametes when a crossover occurs between the interior of an inverted region and its non-inverted homolog (*White, 1973*). As aneuploidy is often fatal in embryos or severely deleterious, this may translate to a fecundity cost for a heterokaryotypic parent: a parent heterozygous for an inversion arrangement. However, depending on a species' reproductive biology, such reproductive fitness effects may be mitigated if the zygotes terminate early or before significant parental resource investment (*Charlesworth, 1994*; *Munasinghe and Brandvain, 2024*). Alternatively, crossovers may be inhibited directly in heterokaryotypes around the inversion by mechanisms similar to crossover interference (*Koury, 2023*), avoided by achiasmy in one sex or all, or aneuploid gametes may be segregated to polar bodies in female meiosis (*Sturtevant, 1921*; *Sturtevant and Beadle, 1936*; *Krimbas and Powell, 1992*; *Coyne et al., 1993*; *Gong et al., 2005*). Regardless of the differences between these scenarios, effectively fewer recombinant offspring are produced in heterokaryotypes.

Inversions may also have direct effects on fitness when disrupting a gene, a regulatory element, or other genomic structure like a topologically associated domain (*Lupiáñez et al., 2015*; *McBroome et al., 2020*). Given that another potential effect of inversions in heterokaryotypic individuals is the deleterious generation of aneuploid gametes, it seems reasonable to expect inversion rearrangements to confer either deleterious or neutral fitness consequences in the absence of association with other variants. However, when a new inversion mutates, it samples a single haplotype from a population, which is likely to have some non-neutral fitness effect, even if mild. Greater natural variation in fitness within a population increases the likelihood of sampling haplotypes with exceptionally deleterious or beneficial associations of alleles, even when there are no other effects on fitness (*Berdan et al., 2023*).

In contrast to the above predictions, a considerable number of well-studied inversion variants are maintained at an intermediate frequency within a population, presumably due to the indirect effect inversions have on reducing recombination and maintaining fit haplotypes that are themselves under selection (reviewed in *Berdan et al., 2023*). The evolutionary forces maintaining both haplotypes are thought to be quite varied between populations and arrangements, but a common first hypothesis for a frequently sampled inversion is that local adaptation in one region or environment is balanced with migration from another (*Kirkpatrick and Barton, 2006*; *Charlesworth and Barton, 2018*). This association may be maintained more strongly when the alleles have beneficial epistatic interactions, and assumed beneficial epistasis between loci is often paired with another mechanism of balancing selection operating at a locus to explain inversion (*Dobzhansky, 1949*; *Dobzhansky, 1950*; *Dobzhansky, 1970*; *Schaeffer et al., 2003*; *Hoffmann and Rieseberg, 2008*). However, unless migration rates are very high, local adaptation models predict inversions to be at high frequency in their favored geographic range but at low frequency elsewhere, and many inversions do not have sampled locations at which they reach very high frequency.

Alternately, chance associations of both an inversion and its standard arrangement counterpart with different recessive deleterious alleles may mimic the fitness profile of a single overdominant locus (sometimes termed associative overdominance), where heterozygotes carrying both main haplotypes experience none of the deleterious effects of a homozygous recessive across the region (*Sturtevant and Mather, 1938*; *Frydenberg, 1963*; *Zhao and Charlesworth, 2016*; *Gilbert et al., 2020*). However, recent theory and simulation suggests that this dynamic is uncommon unless an arrangement polymorphism is maintained at intermediate frequency under some other selective force, and the population scaled recombination rate between arrangements is low (*Charlesworth, 2024*).

Balanced polymorphic loci may be maintained in other ways, under various models and selective pressures. For example, multiple overdominant loci that are each most beneficial as heterozygotes may benefit from linkage maintaining complementary haplotypes (*Sved, 1968*; *Charlesworth and Charlesworth, 1973*; *Charlesworth, 1974*). Several recent studies have proposed that functionally overdominant loci may be more common than previously considered. Specifically, the dominance relation between alleles at a locus may change between different sexes, seasons, life history stages, or other trait or fitness challenge contexts, generally termed 'reversal of dominance', which in some cases can lead heterozygotes to have greatest average fitness (*Rose, 1982*; *Wittmann et al., 2017*;

*Grieshop and Arnqvist, 2018*; *Connallon and Chenoweth, 2019*). Other specific biological dynamics may predict the maintenance of inversion polymorphism in particular ways. For example, inversions may also link drivers and enhancers in meiotic drive systems (*Babcock and Anderson, 1996*; *Mroczek et al., 2006*; *Courret et al., 2019*), and these drive-associated inversions may persist at equilibrium frequencies in the presence of repressors (e.g. *Bastide et al., 2022*).

In natural populations, it can be difficult to distinguish among these alternate hypotheses. They are not always mutually exclusive, and the selective pressures acting upon an inversion may vary across space and time. The frequency patterns of many inversions, including a number in *Drosophila melanogaster*, remain poorly explained.

## Inversions in *Drosophila melanogaster*

*D. melanogaster* is a widely studied organism in many fields of biology, including evolutionary and population genetics. Multiple polymorphic inversions segregate at meaningful frequencies in natural populations of *D. melanogaster* (*Lemeunier and Aulard, 1992*; *Aulard et al., 2002*; *Kapun et al., 2016*; *Kapun and Flatt, 2019*), but even in this model system, it is not entirely clear which population genetic processes are most responsible for shaping their frequencies. In this species, achiasmy in males (*Morgan, 1910*; *John et al., 2016*) and a lack of evidence for female fecundity costs for most inversions (*Sturtevant, 1921*; *Sturtevant and Beadle, 1936*; *Coyne et al., 1993*; *Gong et al., 2005*) suggest that most paracentric inversions in *D. melanogaster* do not generate aneuploid gametes at any appreciable frequency. Further, genome editing tools have allowed for the creation of synthetic inversions, revealing that the tested structural rearrangements are not themselves responsible for the gene regulatory changes that are associated with natural inversions in *D. melanogaster* (*Said et al., 2018*). Instead, given that breakpoints of common *D. melanogaster* inversions occur disproportionately in regions that confer greater recombination suppression (*Corbett-Detig, 2016*), *D. melanogaster* inversions would appear to be maintained primarily for the sake of maintaining linkage associations between distant variants.

*D. melanogaster* originated in southern-central Africa, in seasonally dry Miombo and Mopane woodlands, and (potentially after adapting to a human commensal niche) expanded out across much of Africa and into the Middle East around 13,000 years ago, and into Europe around 1800 years ago (*Sprengelmeyer et al., 2020*). With an ancestral $N_e$ of around 2 million (*Sprengelmeyer et al., 2020*), *D. melanogaster* populations have high levels of standing diversity and high population recombination rates. Many *D. melanogaster* inversions have geographic clines repeated between several colonized regions (*Mettler et al., 1977*; *Reinhardt et al., 2014*; *Kapun et al., 2016*), and inversions often show geographic differences in frequency that exceed genome-wide average differentiation between the same populations (e.g. *Pool et al., 2017*). It has therefore been suggested that local ecological adaptation may shape the distribution of *D. melanogaster* inversions (*Kapun et al., 2016*; *Kapun and Flatt, 2019*). Indeed, it is difficult to explain their geographic differentiation without some type of spatially-varying selective pressures.

Yet, it is unclear whether ecological adaptation is a sufficient explanation for *D. melanogaster* inversion frequency patterns, for two reasons. First, most of the common inversions become less frequent as one samples populations farther from the sub-Saharan ancestral range (or similar warm environments) into potentially more challenging, colder high latitude and altitude environments (*Aulard et al., 2002*; *Kapun et al., 2016*; *Lack et al., 2016a*; *Pool et al., 2017*; *Kapun and Flatt, 2019*) with *In(3 R)Mo* representing an exception to this pattern (*Kapun et al., 2014*; *Kapun et al., 2016*). As these inversions are all derived (being absent from closely related species), local adaptation theory would predict that they arose to protect local adaptation against ongoing migration, more likely in the novel environment if it is experiencing high levels of immigration (*Kirkpatrick and Barton, 2006*). It is unclear why inversions would be needed to adapt to the species' ancestral environments, or why ancestral 'standard' arrangements would have an ecological advantage in derived environments that are very different from the ancestral range.

Second, even in the locations where inversions would hypothetically be ecologically advantageous, they rarely surpass intermediate frequencies (*Lemeunier and Aulard, 1992*; *Pool et al., 2017*; *Kapun and Flatt, 2019*; *Sprengelmeyer et al., 2020*). Under a local adaptation model, that observation would require a secondary explanation that resists fixation, such as very high levels of long distance gene flow or the fixation of recessive deleterious variants on inverted haplotypes (*Ohta and Kimura*,

*1970*; *Zhao and Charlesworth, 2016*; *Gilbert et al., 2020*). While migration may well limit inversion frequencies in some populations, we note that some putatively adaptive Single Nucleotide Polymorphism (SNP) variants do achieve much greater frequency differentials between populations (*da Silva Ribeiro et al., 2022*), and thus we would have to suppose that inversions are under relatively weaker selection to explain their lesser differentiation. Regarding the hypothesis of high recessive load, we note that *D. melanogaster* populations do generally contain substantial recessive genetic load (*Greenberg and Crow, 1960*), and it has been speculated that associative overdominance between sampled backgrounds might boost inversion frequencies in *D. melanogaster* populations during founder events (*Pool et al., 2012*) or in isolated wilderness environments (*Sprengelmeyer et al., 2020*). However, it seems likely that in large human-commensal populations, *D. melanogaster* inversions would escape from linked deleterious variants through gene conversion or double crossover events with standard chromosomes, in light of both the age of most of these inversions (*Corbett-Detig and Hartl, 2012*) and the very high population recombination rates of *D. melanogaster*. Homozygous lines for each common inversion also appear at appreciable frequency in inbred lab strains, demonstrating a lack of perfect linkage of the inversions with recessive lethal or infertile variants (*e.g. Lack et al., 2016a*).

While local adaptation, gene flow, and associative overdominance may all contribute to geographic variation in inversion frequency among *D. melanogaster* populations, it is not clear that these processes are collectively sufficient to explain observed patterns. Alternatively, the allele frequencies and diversity patterns of at least some of these inversions might be primarily shaped by some form of balancing selection acting on inversion-linked variation, in which the balanced frequency is dependent on the environment (*Kapun et al., 2023*). Balancing selection on inversions has received increasing attention in recent literature (*Wellenreuther and Bernatchez, 2018*; *Faria et al., 2019*). In *Drosophila*, inversions have been associated with balancing selection due to seasonal, temporally fluctuating selection (*Machado et al., 2021*). However, the potential for temporally varying selection to maintain stable balanced polymorphisms on its own is somewhat limited (*Gillespie, 1998*). Further, most common inversions maintain relatively high polymorphic frequencies across broad geographic ranges without obvious shared seasonal selective pressures, from tropical and subtropical dry forests to temperate locations with harsh winter conditions (*Lemeunier and Aulard, 1992*). Therefore, we predict that the frequencies of some of these inversions could be in part explained by a mechanism of balancing selection that would operate even within a temporally and spatially homogenous population. Below, we propose such a model in which alleles that contribute to competitive mating display simultaneously contribute antagonistically to survival to reproductive maturity, a tradeoff that offers a mechanism by which the life-history characteristics of a population could generate epistatic balancing selection.

## Sexually antagonistic polymorphism in *D. melanogaster*

Among mechanisms of selection, sexual selection is both prevalent, and provides a simple mechanism for frequency dependence (*O'Donald et al., 1997*). Alleles that benefit an individual in competitive mate choice systems often decrease in fitness as the beneficial allele increases in frequency, as display success depends on the relative quality of a male among local competitors, and the more common a display-enhancing allele is, the fewer competitors it succeeds over. Albeit, the direction and magnitude of the frequency dependent effect can depend on the way mate choice is parameterized even among 'best of n' models, as well as the strength of selective versus stochastic effects (*O'Donald, 1973*; *O'Donald et al., 1997*). Combinations of display-enhancing alleles at multiple loci may similarly gain a disproportionate benefit together, as male mating success in competitive display systems can be skewed towards a few successful individuals (*Bateman, 1948*; *Jones et al., 2002*; *Tatarenkov et al., 2008*, though see *Gowaty et al., 2012*; *Hoquet et al., 2020*). In a mate choice system such as a best of n model in which a female chooses the highest quality male encountered, a marginally higher quality male similarly outcompetes either a marginally lower or an exceptionally worse male. So, the last few alleles to push display into the top quantile should contribute a significantly greater fitness increase to the genotype than the first few display alleles. This interaction of alleles is a form of epistasis – a non-independent fitness interaction between loci – that is emergent and non-specific to the genetic identity of the variants. In other words, the alleles contribute independently to the generation of a trait, and the fitness of the individual is a nonlinear function of that trait (*Blows and Brooks, 2003*; *Rest et al., 2013*; *Otwinowski et al., 2018*). This epistatic interaction suggests a potential for indirect selection on the recombination-suppressing effects of inversions, in order to maintain

combinations of sexually antagonistic variants that have the greatest fitness when carried together. This tendency of selection to favor the reduction of recombination when the haplotypes present are at a fitness optimum is well studied for recombination modifiers including inversions, and has been termed the 'reduction principle' (*Feldman, 1972*; *Feldman and Balkau, 1973*; *Feldman et al., 1980*; *Altenberg and Feldman, 1987*; *Altenberg et al., 2017*; see *Altenberg and Feldman, 1987* for a direct treatment of inversions).

While the emergent epistasis generated by a mate choice system may favor linkage of variants, alleles that contribute only to a sexually selected display are not expected to remain as balanced polymorphisms, rather they are expected to fix, albeit perhaps slowly as their fitness benefits diminish at high frequencies (*Hazel, 1943*; *Dickerson, 1955*; *Charlesworth and Hughes, 2000*). However, if these alleles simultaneously contribute deleteriously to other components of fitness, they are pleiotropic (*Stearns, 2010*), and given the sex-limited benefits of the display they would likely also be sexually antagonistic (*Rowe et al., 2018*), whereby traits increase the fitness of one sex but decrease fitness if found in the other. Pleiotropy can result from traits that are inherently physiologically coupled, from resource allocation restraints, or from the specific role or behavior of genes. Pleiotropy can further be sexually antagonistic when the traits in the tradeoff are differentially important to different sexes, which may be common given the differing fitness requirements of the sexes in a population (*Cox and Calsbeek, 2009*; *Connallon and Clark, 2014*; *Connallon and Chenoweth, 2019*). It should be noted that a model of two pleiotropically determined traits each with potentially sex-specific fitness can be treated separately from a model where one trait has sexually antagonistic fitness effects.

*D. melanogaster* mating involves a male display performance and female acceptance or rejection. A number of experimental evolution studies (*Rice, 1992*; *Rice, 1996*; *Audet et al., 2024*) and genetic studies *Innocenti and Morrow, 2010*; *Cheng and Kirkpatrick, 2016*; *Glaser-Schmitt et al., 2024* have confirmed the prevalence of sexually antagonistic loci in *D. melanogaster*. Pleiotropic alleles that trade off between survival and male display quality represent one clear potential case of antagonism to consider for a model of balancing selection, and one highly relevant to the biology of *D. melanogaster*. Many such antagonistically pleiotropic scenarios are possible under different selective tradeoffs, and indeed some have been proposed for other model inversions with varying mechanisms of maintenance (*Betrán et al., 1998*; *Pearse et al., 2019*; *Mérot et al., 2020*; *Pei et al., 2023*; see Discussion), but we focus on a specific scenario of sexual antagonism here.

Given the relevance of sexual selection and sexual antagonism to *Drosophila* evolution, we propose that polymorphic *D. melanogaster* inversion and standard arrangements may help maintain haplotypes of alleles that specify opposite ends of a sexually antagonistic tradeoff that is subject to balancing selection and for which intermediate states are inherently disfavored. We first explore this model through forward simulation, to establish that it may in theory arise in a population and maintain inversion polymorphism over a significant number of generations. Then we examine the results of an experiment with an outbred laboratory population of *D. melanogaster*, to test whether common inversions show changes in frequency over the course of a generation in a manner consistent with a trade-off between survival and male reproductive success.

## Results

### A model of sexually antagonistic inversions

Here, we consider a population featuring a competitive mate choice system that results in a skewed distribution of reproductive success among males. Within this population, we model a genomic region containing a number of linked loci, each with alternate antagonistic, pleiotropic alleles that trade off between male display success and survival likelihood (of either or both sexes). Critically, we do not directly model any epistatic interactions among loci affecting survival or display traits, nor do we invoke dominance in our diploid model, so any such fitness dynamics emerge from the modeled population processes.

If mate choice is highly competitive, such that each female can choose amongst several or many males, the variance of offspring number among males (*Figure 1A*) may be much higher than for females - yielding a distribution of male reproductive output that follows a relatively convex trend, in which most males produce few offspring. Conversely, superior display quality can enable males to mate successfully much more often than other display phenotypes they compete against (*Figure 1A*).

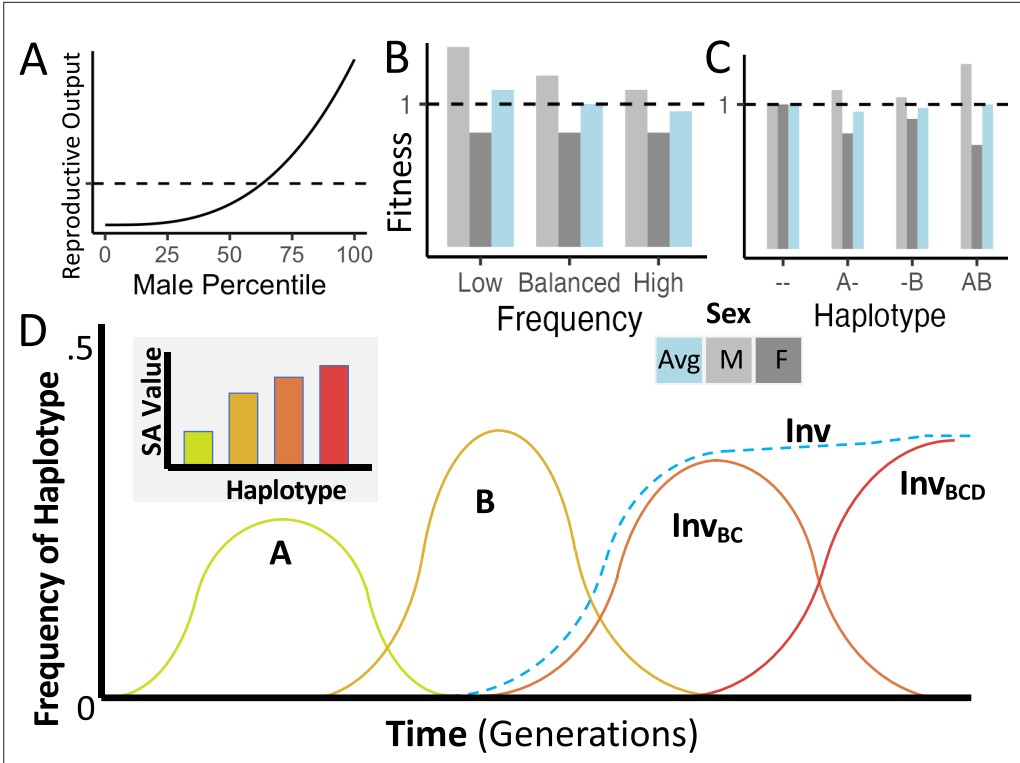

**Figure 1.** Conceptual representations of the proposed model of inversion-associated balanced sexual antagonism. (**A**) Hypothetical skewed distribution of male mating success, yielding a greater variance in reproductive success for males than for females, such as expected under a 'best of n' mate choice model. (**B**) Fitnesses of display-favoring alleles under sexually antagonistic balancing selection, illustrating a pleiotropic variant that should rise in frequency when rare, but decline if frequency exceeds the balanced equilibrium value. (**C**) Fitnesses of display-favoring alleles at two haploid loci under the same model, illustrating synergistic epistasis between displaying-favoring alleles. (**D**) A hypothetical trajectory of four such mutations, in which B outcompetes A, an inversion links B and C to create a more strongly display-favoring haplotype, and then the addition of D to that haplotype furthers the accumulation of antagonistic variants, reaching an equilibrium frequency while displacing less extreme display-favoring haplotypes.

An increase in the display of a male with low-quality results in a smaller change in number of offspring than an increase in display of a male with high quality (change ii in *Figure 1A*), due to the reproductive skew towards highly competitive individuals. So, there is an emergent non-additive fitness effect, where multiple variants increasing the reproductive rank of an individual together have a larger fitness benefit than the combined effects of each substitution separately – a positive, synergistic epistatic effect. This effect occurs whether the difference is from substituting alleles at two loci, or from substituting two homologous alleles at a single locus in a diploid or polyploid organism, which would give an emergent dominance effect. Additionally, fitness conveyed by an allele favoring display quality is also frequency-dependent: since mating success depends on the frequency of comparable display qualities of other males in a best of n model, the relative advantage of a display trait can diminish as more males carry it, for both haploid and diploid populations (*Figure 1B*, male fitnesses; *O'Donald, 1973*; *O'Donald et al., 1997*). If a variant only conveys benefit to mate competition success, it is expected to fix given enough time, as even if its evolution is nearly neutral at high population frequency, selection will push the frequency back up if it drifts down. Whereas, if the variant is antagonistically pleiotropic for a trait not involved in competition between males, such as survival in the face of climate, pathogens, or predators, these fitness costs may typically not be frequency dependent. If the benefit of an antagonistic display-favoring allele overcomes the pleiotropic cost when the variant is rare, its diminished benefit when at higher frequency may reverse its net benefit, while at some intermediate frequency, we predict that these fitness effects should be equal and an equilibrium frequency should be reached and maintained (*Figure 1B*). In the absence of other forces, this dynamic is expected to

maintain the allele under balancing selection indefinitely. We note that this dynamic can be described as a 'negative frequency dependence' of the fitness component for mate choice alone, as the selection coefficient of the favored allele may decrease towards zero while remaining positive, though some authors instead use negative frequency dependence to specifically describe fitness functions in which all alternate alleles are of selective benefit when rare (e.g. *Charlesworth and Charlesworth, 2010*). In addition to the relationship between sex averaged fitness and frequency, the contrasting sex-specific fitnesses of antagonistic alleles under this model could further enhance the potential for balanced polymorphism.

We further predict that multiple such balanced pleiotropic loci should generate interesting emergent behavior. Because of the emergent synergistic epistasis from mate competition (see discussion of *Figure 1A* above), the haplotype with the greatest male display benefit should increase in frequency relative to other haplotypes until reaching an equilibrium at which its relative advantage in mate choice is reduced enough to be balanced by the pleiotropic, frequency-independent, non-epistatic survival cost (*Figure 1B*). The haplotypes and their linked alleles conveying intermediate phenotypes will have less than proportional benefits during mate choice due to the convex relationship between quality and success described above (*Figure 1A*), but a larger, proportional cost to survival against non-intraspecific survival challenges, giving them lower average fitness in the equilibrium state than either of the extreme haplotypes (*Figure 1C*). Therefore, recombination between the two extreme haplotypes will generate less fit haplotypes in the gametes, and a local modifier of recombination that reduces recombination and its associated costs (such as an inversion) will be favored by indirect selection (*Altenberg and Feldman, 1987*; *Otto and Lenormand, 2002*). And so, selection will favor the association of these haplotypes with the alternate standard and inverted karyotypes, which generate fewer recombinant gametes from a heterokaryotypic parent. We further predict that in a population that is close to the balanced equilibrium state for such a pleiotropic trade-off, the population of adults will contain a higher frequency of survival-focused haplotypes than the population of zygotes, but the display-focused haplotypes will increase in frequency in zygotes relative to adults, in line with the life stage-specific fitness differences between these haplotypes.

While *Figure 1* illustrates a haploid scenario for the sake of simplicity, this model extends naturally to a diploid case. The main difference is that heterokaryotypic diploids – those heterozygous for the arrangement – would have intermediate trait values and be less fit than either homokaryotype, for the same selective reasons that an intermediate haplotype with two alleles of opposing effects at different loci is least fit in a haploid context. This is therefore, interestingly, an example of a scenario in which the two most common haplotypes in the population can exhibit underdominant fitness, and yet may still be maintained at a stable balanced equilibrium. By comparison, the simplest models of underdominant fitness predict an unstable equilibrium that disfavors polymorphism (*Pool et al., 2012*, *Charlesworth and Charlesworth, 2010*), a characteristic shared with non-frequency-dependent models of disruptive selection. The cost of generating underdominant offspring might be compensated for by a strong co-directional dominance of the variants involved, such that heterokaryotypic individuals exhibit a phenotype similar to one of the homokaryotypes. However, dominance is not a necessary feature of our model, and it is not invoked in our simulations. Whereas, investigating a wider range of genetic architectures would be a potential direction for future study.

## Simulation results

We investigated the predictions of the above model of inversion-associated balanced sexual antagonism using a purpose-built forward simulation program, 'Sexually Antagonistic Inversion simulator' (SAIsim; see Materials and methods). This simulation program incorporates the generation of new nucleotide and inversion polymorphisms by mutation, where each nucleotide mutation independently draws a positive or negative effect on survival and on male display. It also simulates crossing over and gene conversion and inherently the effects of drift (Materials and methods). SAIsim includes gene conversion between all homologous sites regardless of karyotype, and models crossing over in heterokaryotypic individuals from even numbers of crossover events within the inverted region. It explicitly models mate selection, in that for each female selected to reproduce, a set number of 'encountered' males are randomly selected, and the observed display value of each male for this competitive event is assigned based on the sum of the male's genetic effects and a noise effect resampled at every encounter. The male with the highest observed display value then succeeds in this mating (*Figure 2*).

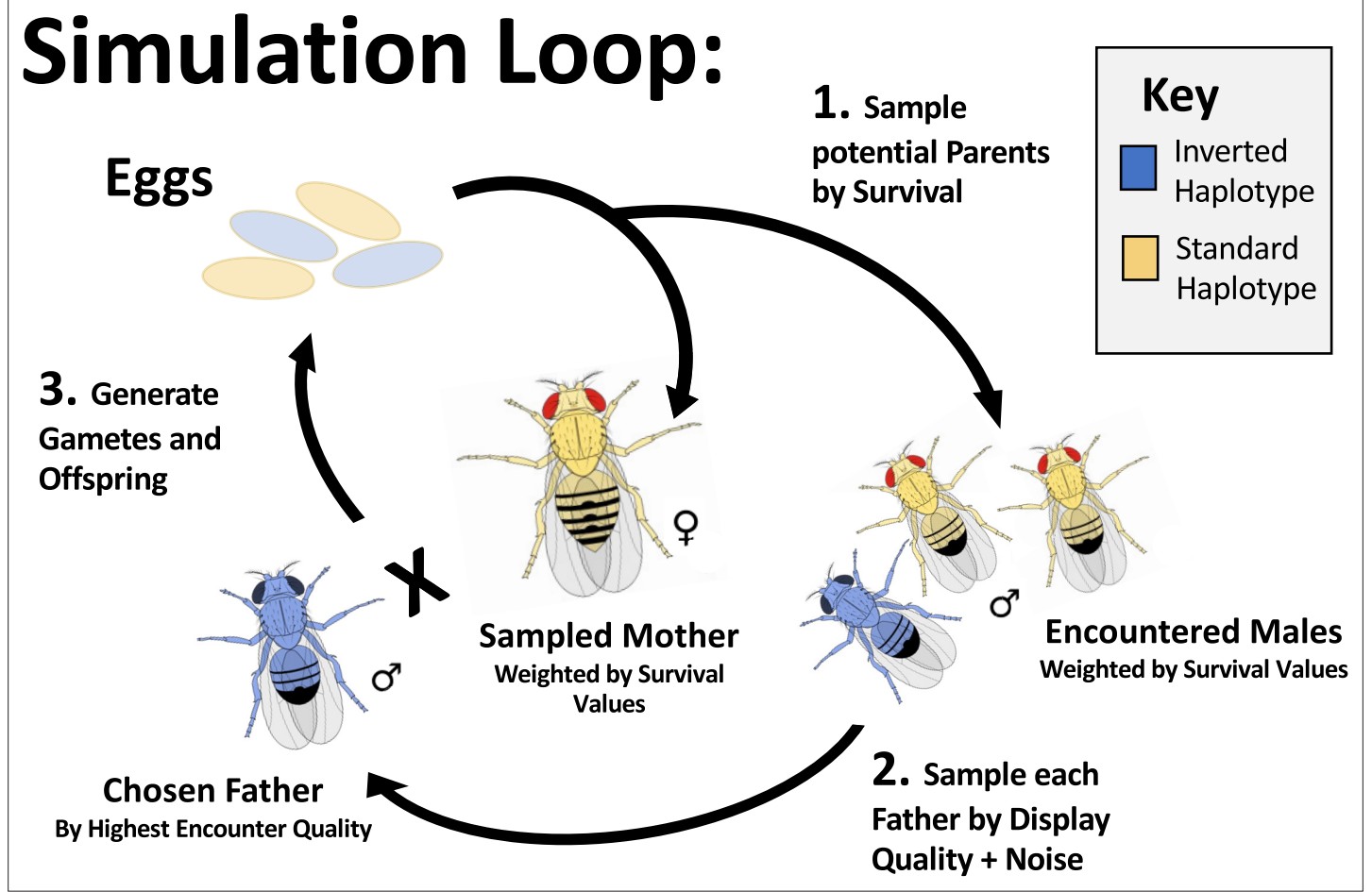

**Figure 2.** The layout of a single simulated generation. Simulated individuals each reach the reproductive stage with probability in proportion to the product of the survival effects of their alleles. To generate each offspring in the next generation, a surviving female is first randomly sampled (with replacement) to be the mother. Then a specified number of surviving males are randomly sampled to generate the pool of males encountered by the sampled female for the best of n mate choice – without replacement for each pool and with replacement between pools. Unless otherwise noted, we defaulted to 100 encountered males in simulations presented here. For each encounter pool, each male is assigned an observed display quality as the sum of the display effects of its alleles, plus a normally distributed noise effect for each encounter. The highest quality male in the encounter pool is selected as the father, and the offspring genome is generated. The female gamete is generated with crossover and gene conversion events. Odd numbers of crossovers in the interior of an inversion are resampled from the same parent, to model the biology of an organism that eliminates aneuploid gametes during oogenesis or exhibits reproductive compensation. *D. melanogaster* males do not cross-over, and so males do not recombine in the simulations presented.

With this simulator, we first confirmed that some individual variants with an antagonistic pleiotropy between survival (initially of both sexes) and male reproduction can generate stable selectively balanced polymorphisms in silico. We simulated a single locus in diploid populations of N=1000, polymorphic for an allele of a specified survival cost and display benefit at an initial frequency of 0.5, with the alternate allele having no trait effect. For each female's mating competition, 100 males were sampled, although see *Figure 3—figure supplement 1* for plots with varying encounter number. After 20 N generations, we calculated the proportion of simulations retaining polymorphism and the average final frequency of the variant across all simulations, revealing that there is a range of paired trait effects for which the mutation consistently remained polymorphic, with intermediate mean frequency, after 20 N generations (*Figure 3*) – implying the action of balancing selection. Alleles with small effect sizes had only a narrow band of selective maintenance, outside of which they were either fixed or lost, while large-effect alleles remained balanced across a larger area of simulated parameter space. While the overall balanced range is somewhat narrow, such variants may persist under balancing selection indefinitely in the absence of other forces, and many small-effect alleles in linkage

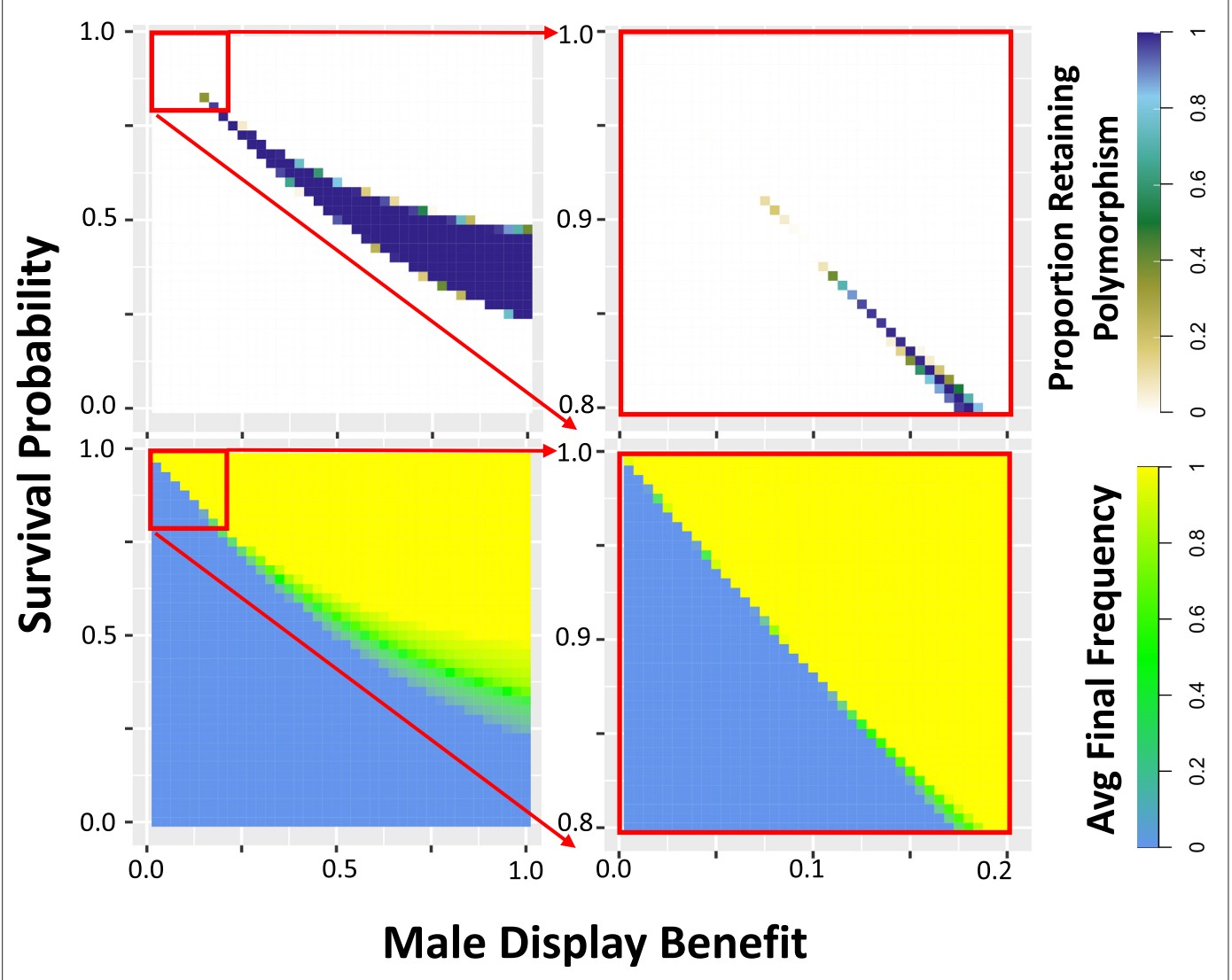

**Figure 3.** Simulations show that a balanced equilibrium frequency exists for certain sexually antagonistic variants. The proportion of simulations retaining polymorphism (top) and the average final frequency (bottom) are plotted for a single locus at which there is one antagonistic allele with the indicated effects reducing survival proportion (of both sexes) and increasing male reproductive quality score, initiated at equal frequency with an alternative allele that has no such effects. Larger values of both effect sizes are plotted in the left panels, with values incrementing by 0.025, and smaller effect sizes (a subset of the larger) are plotted in greater detail in the right panels, with values incrementing by 0.005. Reproductive quality contributes to male success in a best of 100 mate competition with an added normally distributed noise effect of standard deviation 1 (*Figure 2*; Materials and methods). See *Figure 3—figure supplement 1* for plots with sex-specific costs and differing encounter numbers in the best of n mate competition. Most parameters result in either all simulations retaining polymorphism (purple), or else all simulations losing polymorphism (white). However, simulations retaining polymorphism have a broad range of equilibrium frequencies (shades of green). For each parameter combination, 500 replicates were simulated for a diploid population of 1000 individuals evolving for 20,000 generations, with further details as indicated in the Materials and Methods.

The online version of this article includes the following figure supplement(s) for figure 3:

**Figure supplement 1.** Single locus simulations under differing models exhibit shifts in the parameter combinations that produce balancing selection.

blocks may confer larger effects in synergy. In a scenario where only females pay the survival cost, the balanced parameter space of antagonistic trait effects is broader and shifted somewhat (*Figure 3— figure supplement 1*). In a scenario where only males pay the survival cost, and variants are neutral in females, essentially no simulations retain polymorphism (*Figure 3—figure supplement 1*). The number of males sampled to assess mate competition for each offspring also has a significant effect, with larger values giving broader polymorphic ranges, although simulations with costs only in females

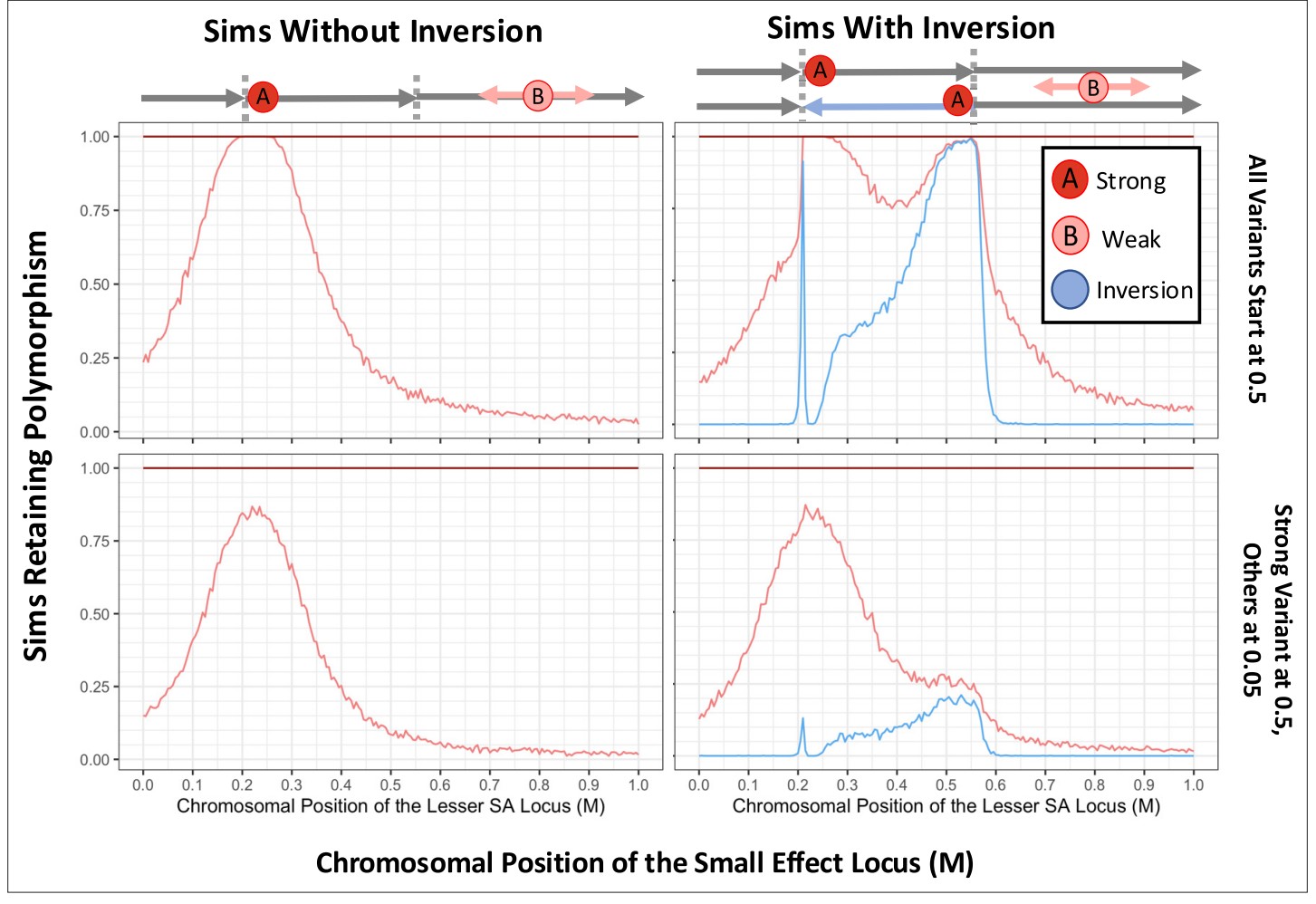

**Figure 4.** Simulated inversions may persist as polymorphisms when linked to sexually antagonistic, pleiotropic variants. In turn, the presence of inversions facilitates the accumulation of antagonistic variation, demonstrating synergistic epistasis. Rates of long-term persistence are shown for simulations with two defined antagonistic variants alone (left panels) and with the additional presence of a defined inversion (right panels). To focus the potential addition of a second linked variant to an antagonistic haplotype, we begin with the stronger variant already at an intermediate frequency in the population (0.5). The weaker antagonistic variant and the inversion either both start at 0.5 frequency as well (upper row) or both start at a lower frequency of 0.05 (lower row), with all variants initially in linkage equilibrium. The 1 Morgan chromosomal segment is diagrammed above the plots, with the inversion breakpoints marked (0.21 M, 0.55 M), chosen to include space across and beyond the inverted region. The proportion of simulations in which the second, variably positioned weaker antagonistic variant (survival and reproductive values 0.86, 0.12) maintains polymorphism is shown (light red line), dependent on the recombination distance from a stronger allele (0.75, 0.3) at fixed position 0.225 M, which always persists (dark red line at 1 across the top of each plot). In the right panels, the proportion of replicates in which the inversion is retained is also plotted (blue line). Values for each parameter combination were averaged over 1000 replicate simulations, for populations of 1000 diploid individuals run for 20 N generations. Gene conversion occurs for each heterozygous variant at a probability of $10^{-2}$ per female meiosis with random direction. Crossover arrangements generating aneuploidy, where there are an odd number of crossovers within the inverted region of a heterokaryotypic parent, are resampled from the same parent to represent removal to polar bodies or reproductive compensation.

The online version of this article includes the following figure supplement(s) for figure 4:

**Figure supplement 1.** After 20 N generations of simulation, populations approach an equilibrium in which the haplotypes with both display-favoring or both survival-favoring variants are most fit, and selection removes recombinant haplotypes, maintaining high linkage.

show robust polymorphism even when only sampling two males for each competition (*Figure 3— figure supplement 1*).

We then tested the behavior of two linked polymorphic loci with antagonistic alleles. We selected antagonistic effects for these alleles from variants previously found to undergo balancing selection separately, and performed simulations with varying genetic distances between them (*Figure 4*). Crossing over and gene conversion were simulated, with gene conversion occurring for each

heterozygous variant at a probability of $10^{-2}$ per female meiosis per site, with random direction. This behavior was intended to be a conservative approximation of the number of gene conversion events affecting a given site in *D. melanogaster*, scaled to simulations of N=1000 (*Comeron et al., 2012*). The stronger antagonistic variant (modelled at fixed position 0.225 M) persisted in all simulations, in line with expectations, and so we focus on whether the weaker variant (at varying positions) also persisted with it. As seen in the upper panel of *Figure 4*, the weaker variant (pale red line) was retained at close linkage, as the haplotype containing both antagonistic variants acts as a single larger effect supergene. The haplotypes with intermediate trait values were less fit, leaving less offspring per haplotype across simulations (*Figure 4—figure supplement 1A*), whereas the extreme haplotypes were maintained at much higher frequency (*Figure 4—figure supplement 1B*), generating linkage disequilibrium between display-favoring variants ($r^2$=0.2139 across simulations in which the smaller effect variant was positioned at 0.250 M). This demonstrates the synergistic epistasis proposed in our verbal model. The maintenance of haplotypes AB and ab under selection while Ab and aB are selectively removed relies on the lesser relative fitness of Ab and aB. And since survival values are multiplicative, this additional contribution must come from the mate success of AB being disproportionately larger than Ab or aB. As the weaker variant was then moved farther from the stronger variant, recombination increasingly generated unfit intermediate genotypes compared to the extremes of the reproduction-survival tradeoff, and the stronger variant outcompeted the weaker variant among these haplotypes despite the epistatic benefit from associating the two variants together, leading to the loss of the weaker variant. In the scenario examined by *Figure 4—figure supplement 1*, the differences among haplotypes in sex-averaged fitness are modest, whereas the sex-specific fitness effects are more dramatic. These results could indicate that in addition to the frequency-dependent dynamics proposed above, the contrasting fitness effects in females and males may also play an important role in maintaining balanced polymorphism, particularly when near frequency-dependent balance (*Connallon and Clark, 2014*; *Kaufmann et al., 2023*).

We then examined the same spacings between antagonistic variants across a region containing a polymorphic inversion, with its left breakpoint close to the stronger variant, and the right breakpoint within the range of the positions in which the weaker variant was simulated (*Figure 4*, lower panel). When the inversion was added, the weaker variant was then retained when the two antagonistic variants were relatively more distant, due to association of the weaker variant with the right-hand inversion breakpoint (pale red line), at which point this association also preserved the inversion itself in a polymorphic state (light blue line). While less predictable, a weaker antagonistic variant near the middle of the inversion also appeared to maintain polymorphism more frequently in systems starting with an inversion than without, that is across approximate positions 0.30–0.45 M in *Figure 4*, in spite of its weaker linkage with the inversion itself due to double cross-over events. The inversion was not maintained when the weaker antagonistic variant was within the inversion but tightly linked to the stronger variant (see positions 0.22–0.25 M); here there is little crossing-over for an inversion to suppress, and so the effect of the inversion approaches neutrality. Interestingly, there was a small window of weaker variant position in which the inversion remained polymorphic despite the already tight linkage between SA variants, where the inversion seemingly does not modify recombination much (see positions 0.2–0.22 M). Here, the weaker antagonistic variant is more tightly linked to the breakpoint than to the stronger variant in this range, and presumably benefits more consistently from the crossover reduction. The maintenance of the weaker variant drops more rapidly outside of the inversion than over similar distances in the simulation without inversions (positions 0–0.2 M, bottom versus top panel). This seemingly counter-intuitive result may stem from crossover events that result in aneuploid gametes being resampled in our simulations, which effectively increases the crossover rate outside the inversion breakpoints, a behavior which should be biologically reasonable in taxa with aneuploid chromosome elimination or reproductive compensation. So, for a weaker variant outside the left inversion breakpoint, the presence of the inversion can increase the crossover rate between the two antagonistic variants, and thus reduce the probability of the weaker variant surviving by retaining linkage to the stronger one.

In further simulations, we began with a population initially free of variation, in which sexually antagonistic mutations arise stochastically. To assess differences between longer or shorter antagonistic regions, or higher or lower crossover rates, we modeled populations with parameters scaled to either 1 Mb or 10 kb, using estimates from *Comeron et al., 2012* and *Huang et al., 2016*. The mutations

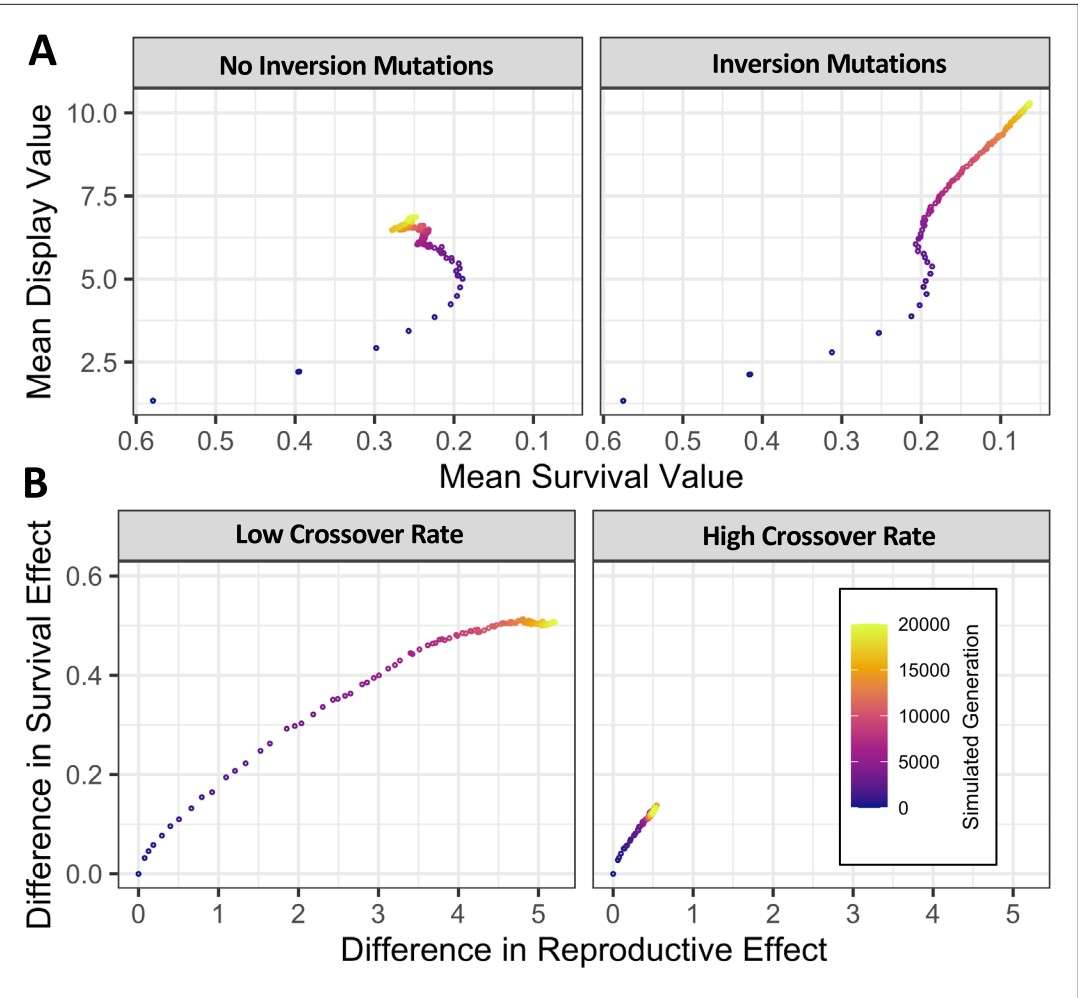

**Figure 5.** Simulated populations with stochastic mutation of new inversions and sexually antagonistic variants generate persistent inversion polymorphisms and accumulate linked sexually antagonistic variants. (**A**) In simulated populations with stochastic mutation of new variants having pleiotropic effects on survival and male display, levels of antagonism increased with time before stabilizing at an intermediate value (left panel). Whereas simulations which also allowed the mutation of new inversions continued accumulating stronger antagonism throughout the simulated time interval (right panel). (**B**) In simulations which allowed the mutation of new inversions, a low crossover rate (left panel, 43.6 cM in females after scaling) allowed populations to accumulate larger differences between the most common karyotype and all others than when the rate of crossover was particularly high (right panel, 43.6 M in females after scaling). Gene conversion occurs at each heterozygous variant with a scaled probability of 0.1295 per heterozygous variant in both scenarios. Values given are the mean of 1000 replicate simulations of 1000 diploid individuals run for 20 N generations.

had pleiotropic effects increasing reproductive quality score and decreasing survival proportion, each drawn sampled independently (Materials and methods). Depending on the effect scores drawn, a mutation could be broadly advantageous (potentially leading to a selective sweep), broadly deleterious, or potentially subject to balancing selection (*Figure 3*, *Figure 3—figure supplement 1*). These mutations arose at a rate equal to one thousandth of the scaled single nucleotide mutation rate given in *Huang et al., 2016*. Inversions (of random lengths and positions) also arose stochastically, at a hundredth of the rate of antagonistic variants. The frequency trajectories of antagonistic variants and inversions were tracked, and the simulation was run for 20 N generations to allow near-equilibrium patterns to be established. These simulations demonstrated the potential of a population without initial genetic variation to establish alternate karyotypes which link sexually antagonistic, pleiotropic alleles and an inversion, to maintain polymorphism in a balanced manner, and then to subsequently accumulate significant antagonistic character (*Figure 5A*), although the degree of antagonism

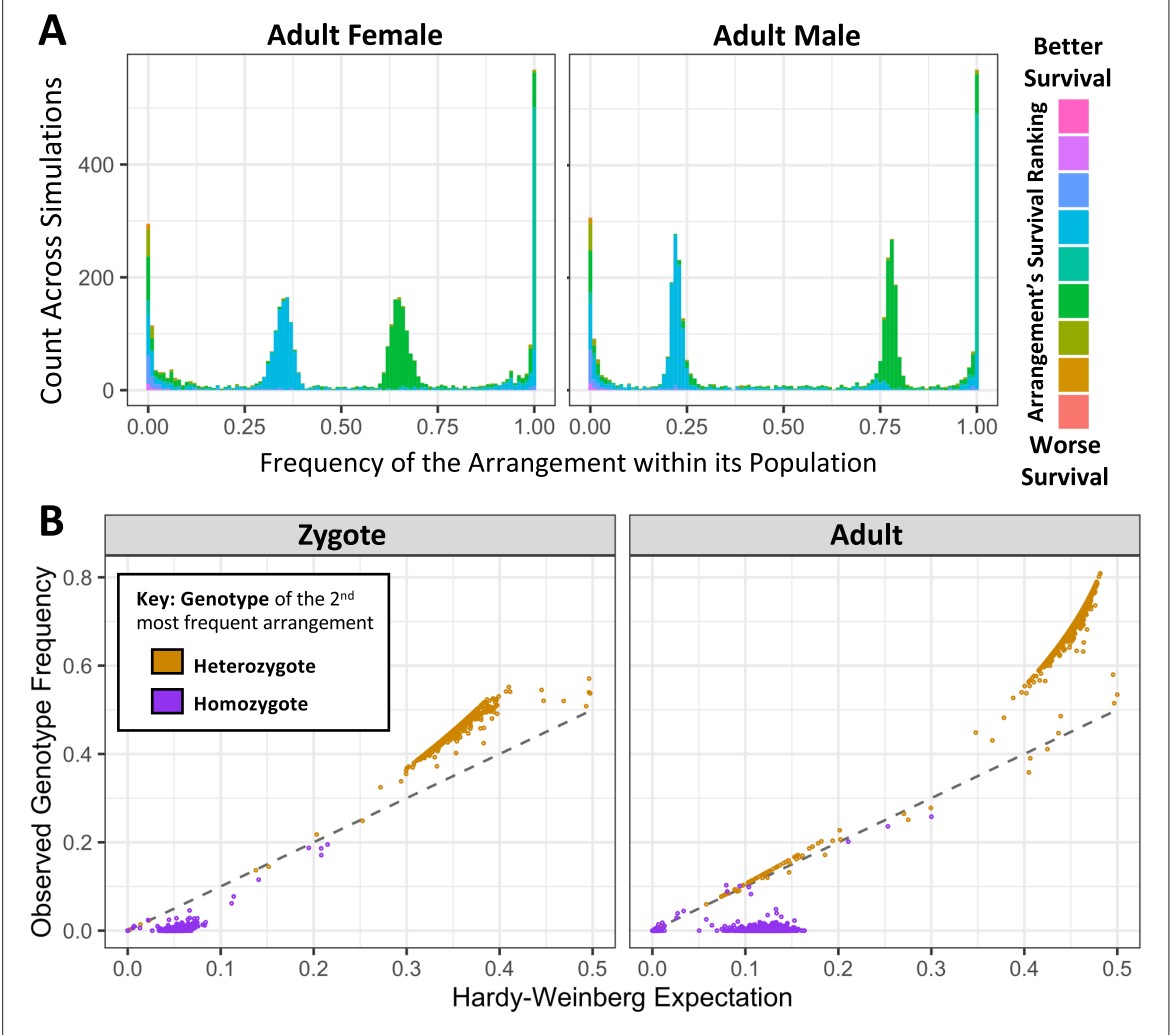

**Figure 6.** Antagonism-associated karyotypes reach predictable equilibrium frequencies in simulations with antagonistic and inversion mutations, when survival costs are shared between sexes. (**A**) Histograms of the number of karyotypic arrangements at a specified frequency across all simulation replicates in females (left) and males (right), colored by how the average survival effect of that arrangement ranks compared to the others in its population. Ranks are normalized to be relative to the median arrangement value, to better account for rare arrangements and ties. These simulations reach an equilibrium with two predominant arrangements, with the more survival-focused arrangement around an overall frequency of 0.25 and more common in females, and the more male-competition-focused arrangement around a frequency of 0.75 and more common in males. (**B**) Because successful males tend to be homozygotes and successful females heterozygotes, the less frequent of the two major arrangements (which tends to favor survival) has homokaryotype frequencies far below the Hardy-Weinberg expectation, and there is an excess of heterokaryotypes. These genotype frequencies shift between zygote (left panel) and adult (right panel) populations as the genotype affects survival.

The online version of this article includes the following figure supplement(s) for figure 6:

**Figure supplement 1.** A histogram of the observed inversion frequencies in simulations with mutational generation of inversions with random position and length.

accumulated depends strongly on the rate of crossing over (relative to the occurrence of antagonistic mutations; *Figure 5B*).

The simulations that successfully maintained balanced antagonistic chromosomes also had an interesting distribution of allele and genotype frequencies of the balanced haplotypes at equilibrium. Without any selection, the inversion frequencies followed a pattern characteristic of the expected neutral site frequency spectrum (*Figure 6—figure supplement 1*). But strikingly, in parameter spaces where simulations with shared survival costs between sexes maintained polymorphism under balancing selection, inversion frequency distributions were tightly clustered around 0.25 and 0.75 among the zygote population (*Figure 6*, *Figure 6—figure supplement 1*). We found that this outcome was

related to the genotype frequencies of reproducing adults. These simulations evolve toward strong antagonism, and it is predominantly males homozygous for the display-favoring haplotype (whether that is inverted or standard) who succeed in reproducing, and these males therefore contribute one copy of the display-favoring haplotype to each of their male and female offspring. However, this haplotype's significant cost to viability and longevity would select for the heterokaryotypic females over the display-favoring homokaryotypes. Given that successful parents would predominantly be heterokaryotypic females and display-favoring homokaryotypic males, around three quarters of trans-mitted chromosomes have the display-favoring arrangement and one quarter does not, yielding the 0.25 and 0.75 zygote inversion frequencies seen in simulations. Although these are autosomal simu-lations, selection yields an inheritance pattern similar to that of a ZW sex chromosome system, where the Z chromosome is homokaryotypic in the fathers, and heterokaryotypic with the W chromosome in the mothers.

The sex-specific fitness dynamics present at equilibrium under this model also resulted in population-wide genotype frequencies that depart from standard Hardy-Weinberg expectations. If

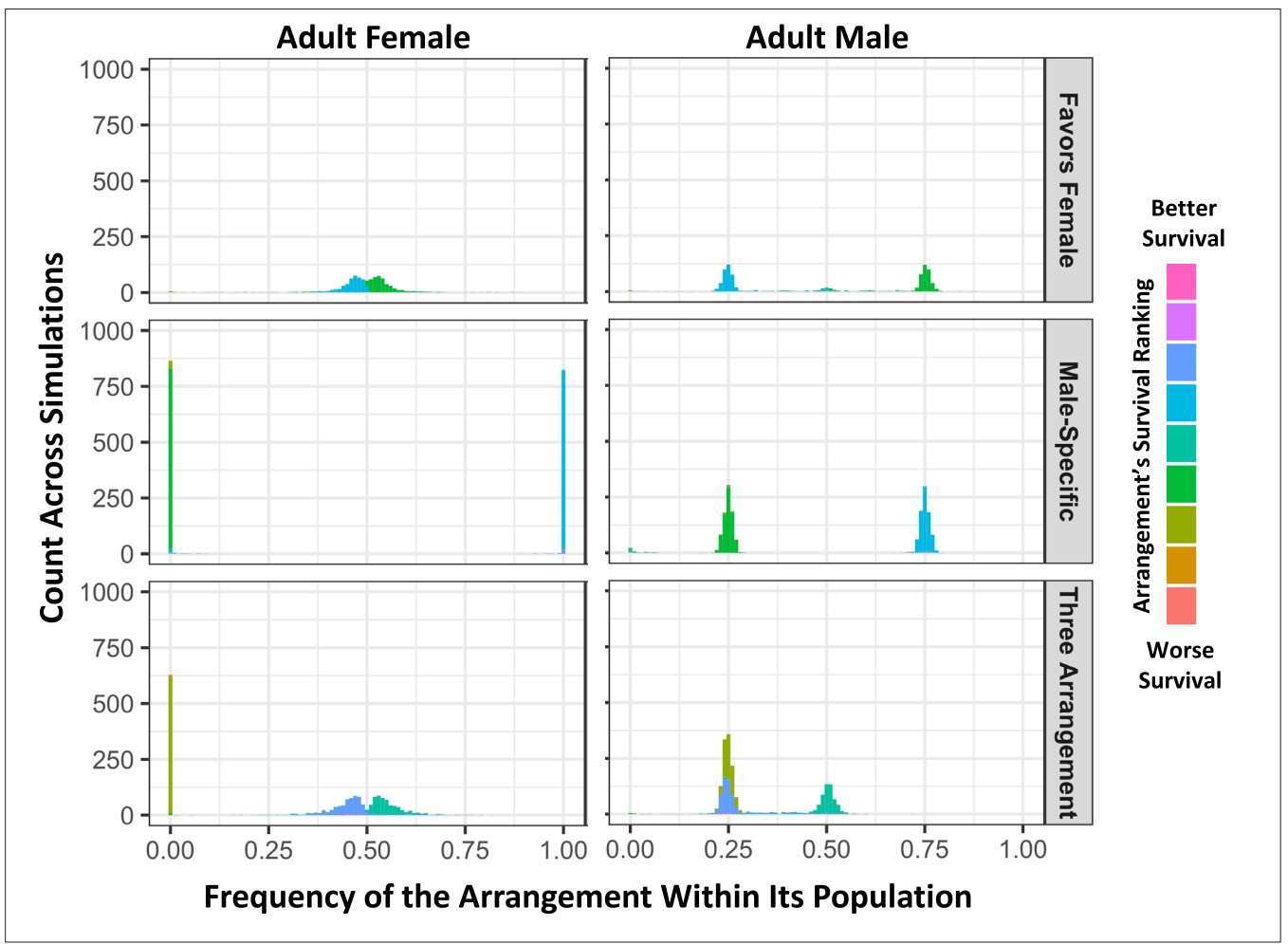

**Figure 7.** In simulations with female-limited survival costs, three distinct outcomes are observed involving either two or three balanced haplotypes. From simulations with randomly occurring antagonistic and inversion mutations, histograms of inversion frequencies across simulations for surviving females (left) and males (right) are partitioned into three categories based on sex-specific haplotype frequency outcomes. The 'Favors Female' category (top) includes simulations in which there are exactly two arrangements with frequencies between 0.1 and 0.9 in both sexes (20.0% of replicates). The 'Male-Specific' category (middle) includes simulations in which there are two arrangements with whole-population frequencies between 0.1 and 0.9, but one arrangement is absent in adult females (37.0% of replicates). The 'Three Arrangement' category (bottom) includes simulations in which there are exactly three arrangements with whole-population frequencies between 0.1 and 0.9 frequency (40.6% of replicates). 2.3% of simulations did not fit in these categories and had only one predominant arrangement. As in *Figure 6*, inversions are colored by how the average survival effect of that arrangement ranks compared to the others in its population, and ranks are normalized to be relative to the median arrangement value, to better account for rare arrangements and ties.

the preponderance of matings are between heterokaryotypic females and homokaryotypic display-favoring males, then homokaryotypes for the survival-focused haplotype will rarely be produced (as confirmed in *Figure 6B*). We note that in cases where the standard arrangement favors male display, then especially if antagonism is strong, this model could explain an observed deficiency or absence of inversion homokaryotypes, even if the inverted arrangement was not associated with any recessive deleterious load. Further, heterokaryotypes may become disproportionately frequent after viability selection (*Figure 6B*). And still, this enrichment for heterokaryotypes reflects neither the existence of recessive deleterious load associated with each arrangement, nor overdominance with respect to viability. Hence, our model may offer an alternative explanation for deviations from Hardy-Weinberg genotype frequencies observed for polymorphic inversions in natural populations.

We further ran simulations in which the survival costs were not shared, but instead specific to females, representing a distinct, stronger form of sexual antagonism. In these simulations, we surprisingly recovered three different equilibrium arrangement states (*Figure 7*). In 20.0% of simulations, we observed a state similar to the shared costs simulations presented in *Figure 6*, where we infer that fathers were homokaryotypic for a display-favoring arrangement, and mothers were heterokaryotypic and carried survival-focused arrangement that is observed in surviving but not reproducing males. For 37.0% of simulations, there were again two predominant arrangements, but with transmission patterns similar to an XY sex chromosome system rather than a ZW sex chromosome system. In these cases, we infer that the mothers were homokaryotypic for an X-like survival-focused arrangement, but the fathers were heterokaryotypic and carried a Y-like display-focused arrangement that was apparently lethal in females. In 40.6% of simulations, we instead detected three intermediate frequency haplotypes: with both Y-like and W-like arrangements restricted to reproducing fathers and mothers respectively, and a third shared arrangement with intermediate phenotypic effects (*Figure 7*).

Specific properties of our simulations may have influenced the occurrence of these three stable states under the female-limited cost model. To reduce the computational complexity of both crossovers and arrangement-specific trait values, the simulations do not allow two inversions on the same individual chromosome, whether overlapping or not, although two chromosomes sampled from the population may have distinct inversions with overlapping breakpoints. This means that if two

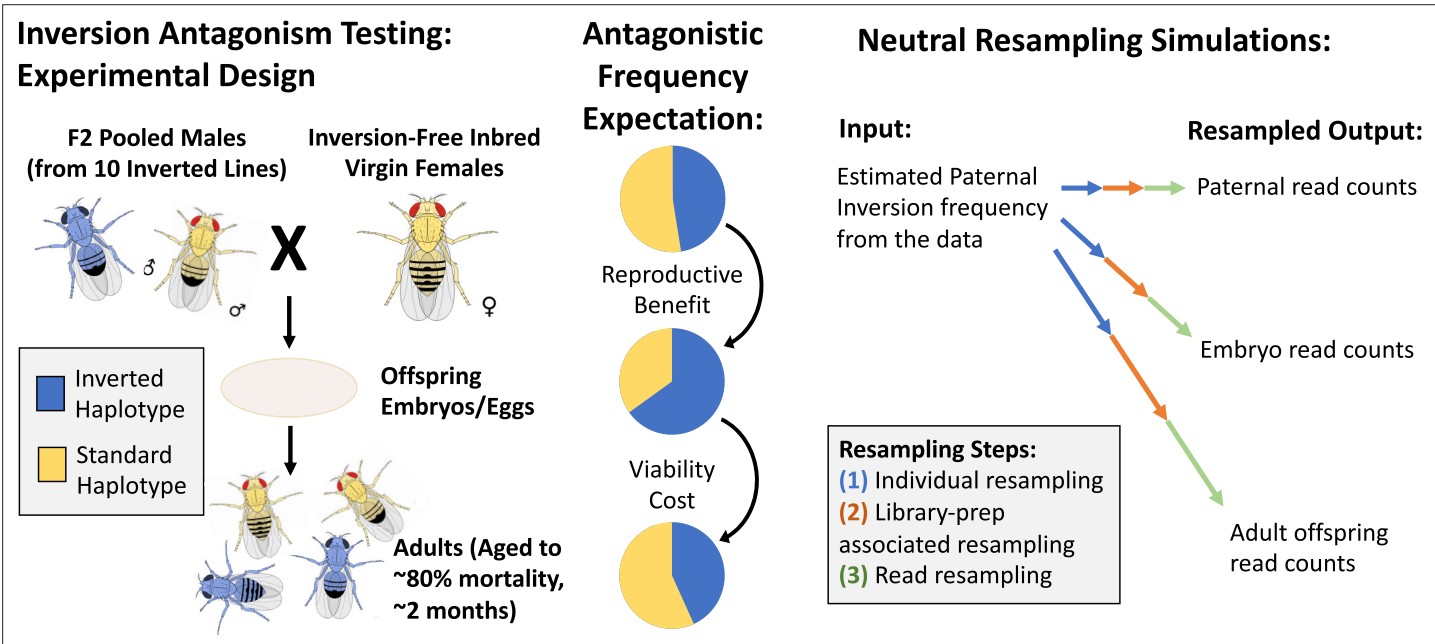

**Figure 8.** The layout and potential expectations of the laboratory evolution experiment. We cross outbred males from a high-inversion population to inbred, non-inverted females from a specific inbred line, and collect DNA samples from the fathers, embryo offspring, and aged adult offspring for inversion frequency estimation via sequencing (left). We hypothesize from our antagonistic pleiotropy model that one or more of the four inversions present on separate chromosome arms in our experimental populations may experience opposing selection between fathers and embryos versus between embryos and aged adults (middle). To test the significance of observed frequency changes, we designed a resampling model to represent the sources of neutral variation expected in the experiment (right).

arrangements, each with an inversion, become the prevalent haplotypes and thereby remove all standard arrangement chromosomes from the population, there is no mutational path to generate a third arrangement. However, a portion of the two-state simulations still had the ancestral, non-inverted arrangement as one of the two haplotypes present, which could have been mutated to a third arrangement to establish the three-arrangement pattern. We are not sure which selective or mutational barriers may prevent transitioning between each state once established. One clear barrier is that it is difficult to generate a less extreme arrangement from a Y-like arrangement, as these can accumulate an indefinite number of mutations since males experience no costs in this set of simulations.

## Empirical tests for inversion-associated tradeoffs

A specific motivation for developing and evaluating the model of inversion-associated, balanced sexual antagonism was to potentially better explain the pattern of inversion frequencies observed in *D. melanogaster*, in which multiple inversions reach some of their highest frequencies in populations from the ancestral range or with similarly tropical/subtropical climates. We therefore set out to test experimentally whether any of four inversions common in a Zambia population of this species show evidence for tradeoffs between male reproductive success and survival (in terms of viability and/or longevity).

We assembled an outbred population with moderate frequencies of each inversion to obtain karyo-typically diverse males. We set up four crosses using these males along with females from one of four different inversion-free inbred Zambia lines (*Figure 8*). Each of these parental crosses occurred in a single large population cage, with populations of at least 600 and equal numbers of each sex (*Supplementary file 1a-b*). We developed a PCR and next-generation sequencing-based assay to estimate inversion-associated SNP frequencies in each pool of males and in each corresponding pool of embryos (see Materials and methods; amplicons and primers given in *Supplementary file 1c*). These SNPs were based on our own analysis of variants that showed perfect association with inversions among 197 Zambia genomes (given in *Supplementary file 1d*; see Materials and methods). We then implemented a resampling method to test for significant inversion frequency changes between parents and offspring (while accounting for the variance observed in technical replicates; see Materials and methods), in order to determine whether males carrying each inversion were more or less successful in reproducing than the average male. We also compared frequencies between embryos and aged adults collected 10 weeks after eclosion, to investigate the relationship between inversions and viability/longevity. These aged adults eclosed either in the first or second half of the collection period from the same batch of eggs, thus allowing us to test for inversion associations with development time. We calculated p-values from the resampling tests to assess the significance of each individual change in inversion frequency between relevant samples (*Supplementary file 1e-h*). However, testing our primary hypothesis that a given inversion had opposing influences on survival and male reproduction, which would provide evidence consistent with our model of balanced sexual antagonism, requires considering the joint event of observing two opposing frequency changes. We therefore calculated p-values from resampling tests designed to account for the dependence between the two changes (see Materials and methods; p-values given in *Supplementary file 1i-j*), and we focus on these results below.

When we initially examined the net effects of male reproduction and survival, we found that in a majority of cases (in 13 out of 16 inversion-line trials; binomial p=0.02; *Figure 9*), a given inversion increased in frequency in aged adult offspring compared to the null expectation based on the frequency in fathers. The remaining three cases were all for *In(3 L)Ok*. Ten of the 13 increases were primarily due to increase in frequency between embryo and aged adult samples, suggesting that in terms of viability and/or longevity, inversion heterokaryotypes often had higher fitness than standard homokaryotypes (no offspring were homokaryotypic for an inversion under our crossing design). We note that even standard homokaryotypes among the fathers and offspring should not have broad-scale homozygosity within the inverted region, but should instead be heterozygous for unique standard-arrangement haplotypes from two independent Zambia strains, and that long genomic tracts of identity-by-descent between Zambia strains are quite rare in this large, ancestral-range population (*Lack et al., 2016a*). Hence, we do not expect that inversion heterokaryotypy is buffering against deleterious effects of inbreeding. Instead, this fitness advantage suggests either an advantage

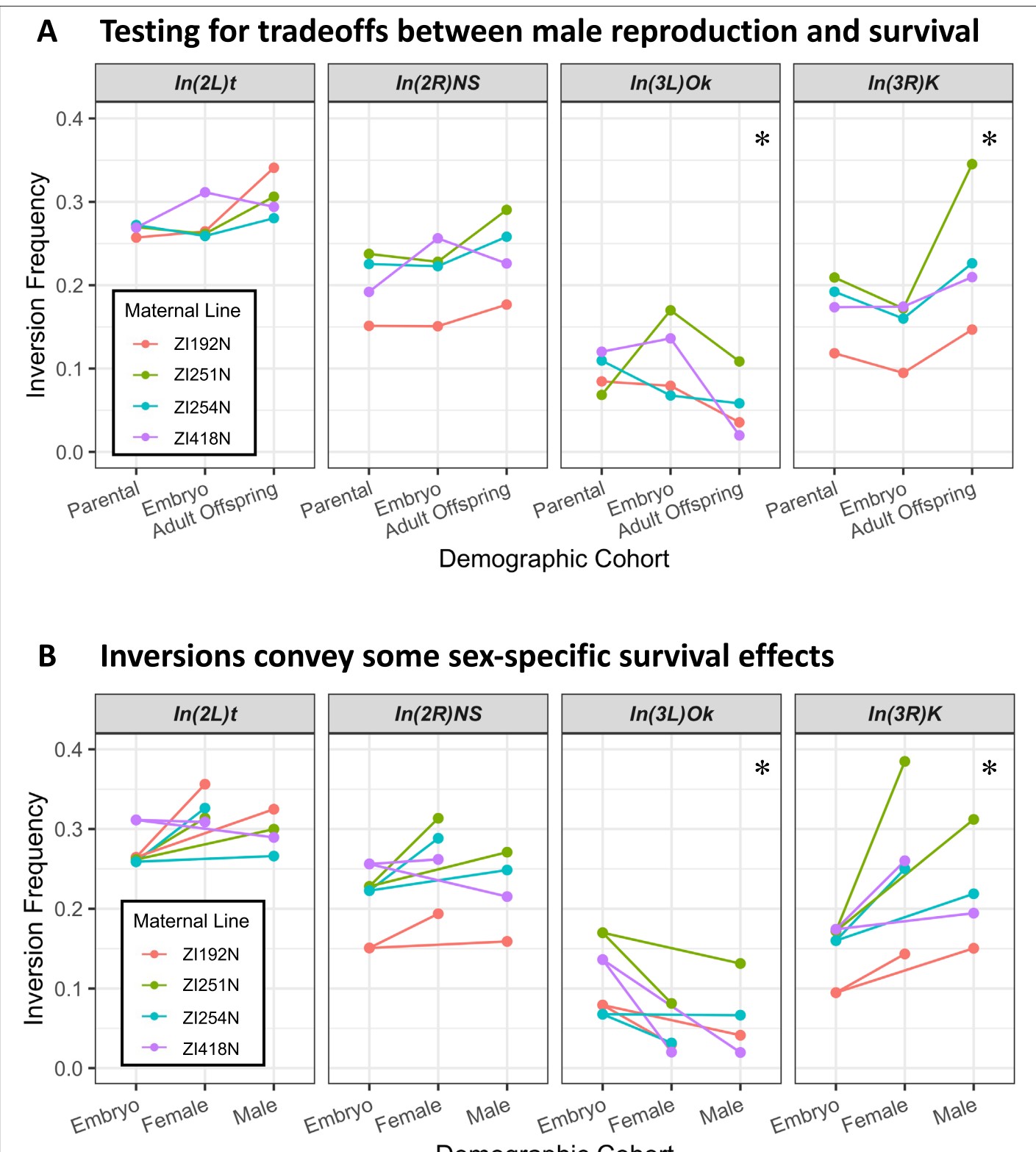

**Figure 9.** Some *Drosophila* inversions show evidence for antagonistic tradeoffs and sex-specific survival. (**A**) Testing for antagonistic fitness effects on reproduction and survival (encompassing viability and longevity). Each inversion's frequency in parents after collection (2–3 weeks from eclosion), embryo offspring, and aged adult offspring (10 weeks from eclosion) are shown for each maternal inbred line. Parental frequency is represented instead of paternal frequency for easier visual comparison, and is taken as the paternal frequency divided by two to account for the inversion-free inbred

*Figure 9 continued on next page*

Figure 9 continued

mothers. Adult offspring frequencies are taken from the sum of estimated allele counts across all adult offspring cohorts. * indicates p-value <0.05 for evidence of significant reversed frequency change from parental to embryo and from embryo to offspring, combined across all four maternal line crosses and corrected for multiple tests, when compared to neutral simulated experiments (details in Materials and methods). (B) Testing for evidence of sex-specific survival effects. The changes in inversion frequencies between embryos and the aged adult offspring are again illustrated, now separated by their sex. * indicates p-value <0.05 for significant differences in female versus male inversion frequency among adult offspring, combined across all four maternal crosses and corrected for multiple tests.

The online version of this article includes the following figure supplement(s) for figure 9:

**Figure supplement 1.** A plot of the observed inversion frequencies in early-eclosing and late-eclosing adult offspring cohorts, compared to the embryo frequencies.

for heterokaryotypy in particular, or perhaps more likely, that multiple inversions harbor alleles that benefit survival or longevity under our experimental conditions (see Discussion).

When we separated the effects of male reproduction (based on father to embryo changes) and survival (embryos versus adults), we found that the effects of inversions on survival were generally consistent across maternal genotypes. Three inversions were associated with increased survival to 10 weeks: *In(3 R)K* was in all four crosses, while *In(2 L)t* and *In(2 R)NS* each showed elevated survival with all maternal strains except ZI418N (*Figure 9A*). In contrast, *In(3 L)Ok* was associated with lower survival in each cross (*Figure 9A*). With regard to male reproduction, the trajectory of inversion frequencies from fathers to embryos to offspring was less consistent between different crosses (*Figure 9A*), implying that differences in male reproductive success may be dependent on the maternal genotype, which is consistent with past findings in *D. melanogaster* (*Reinhart et al., 2015*). Despite such variability, each of the four inversions had multiple frequency changes that departed strongly from null expectations (*Figure 9A*; *Supplementary file 1e-j*), as might be expected if selection was present but varied in strength and direction between crosses and life stages.

To test our central hypothesis of a tradeoff between male reproductive success and cumulative survival during development and adult aging, we used neutral resampling to test the likelihood of observing the magnitudes of opposing inversion frequency changes (between fathers and embryos, and between embryos and aged adult offspring) under the null hypothesis, integrating across the four maternal crosses and accounting for multiple test correction (Materials and methods). For *In(3 R)K*, which had particularly consistent changes in frequency across replicate lines (decreasing between parents and embryos, and then increasing between embryos and aged offspring; *Figure 9A*), this frequency reversal was significantly non-random (p=2.70 × 10⁻³; *Supplementary file 1k-l*). In contrast to *In(3 R)K*, *In(3 L)Ok* was associated with (variably) higher male reproductive success and (consistently) lower survival; this reversal was also statistically significant (p=2.97 × 10⁻⁷).

We also tested for frequency differences between aged adult offspring of different sexes and from early-eclosing versus late-eclosing cohorts. All four inversions displayed differences in frequency between female versus male offspring (*Figure 9B*), consistent with differential effects on sex-specific survival. Strikingly, *In(3 L)Ok* and *In(3 R)K* each displayed sex survival differences in alignment with their antagonistic effects implied above: *In(3 L)Ok* (which favored male reproduction) reduced female survival more strongly than male survival (p=0.0450), whereas *In(3 R)K* (associated with lower male reproduction) favored the survival of females more than it did males (p=4.61 × 10⁻³; *Supplementary file 1m*; see Materials and methods). The other two inversions, *In(2 L)t* and *In(2 R)NS*, each consistently favored female over male survival in all four maternal crosses, although their raw unidirectional cross-combined p values (0.0260 and 0.0126) were not significant after multiple test correction (p=0.282 and 0.137). In contrast, we found no consistent differences in inversion frequency between early- and late-eclosing offspring (*Figure 9—figure supplement 1*, *Supplementary file 1n*).

Estimated selection coefficients for each tested change based on a Wright-Fisher population approximation are available in *Supplementary file 1o*; however, these may overestimate inversion fitness effects since the survival terms reflect the probability of reaching advanced age. Further, there may be an influence of differential sex-specific mortality during aging leading to greater frequency shifts in females. We collected a total of 1960 females and 3156 males, and if the survivors represent 20% of an original population with an equal sex ratio, then there was 84.7% female mortality and 75.3% male mortality. Additionally, there was some mortality among the parents over the laying period, so some of the frequency change observed in embryos may be attributable to paternal

mortality. The mean paternal mortality was 9.83% among the crosses, with a range of 5.11–15.77% mortality (*Supplementary file 1b*), and the fathers were at most 3 weeks old, suggesting mild effects on inversion frequency estimation. Overall, we conclude that *D. melanogaster* inversions have distinct and often context-dependent influences on male reproduction and on survival, with some evidence for antagonistic tradeoffs of the sort proposed in our model.

## Discussion

Here, we have introduced a new conceptual model in which (1) sexually antagonistic variation is present and subject to balancing selection that maintains extreme genotypes favoring either male reproduction or survival, and (2) inversions are predicted to facilitate the aggregation of progressively more antagonistically differentiated haplotypes. We have tested this model using simulations, confirming the predicted efficacy of balancing selection and demonstrating that inversions indeed facilitate the construction of haplotypes containing multiple antagonistic variants. Finally, we have deployed *Drosophila* lab experiments to uncover some evidence for inversion-associated tradeoffs between reproduction and survival, depending on the specific inversion and strains tested.

### Inversions in *D. melanogaster*

We have presented results from a single generation laboratory evolution experiment, in which we crossed outbred male *D. melanogaster* carrying inversions *In(2 L)t*, *In(2 R)NS*, *In(3 L)Ok*, and *In(3 R)K* to four separate inbred female lines and tracked the frequency of the inversions in the initial males, embryonic offspring, and aged adults. For most inversions, the strength and direction of the tradeoff varied notably between crosses. This might suggest high variance in their phenotypic effects or potentially maternally dependent variation in male reproductive-success, but is not distinguishable from experimental noise with only one replicate per maternal line. However, greater consistency was observed for *In(3 R)K*, which was associated with lesser male reproductive success but greater survival. A significant tradeoff was also inferred for *In(3 L)Ok*, which favored male reproduction (albeit variably) over survival. These results provide significant but measured evidence for the presence of sexually antagonistic autosomal inversions in this model species, while emphasizing the importance of genetic background.

As indicated in the Introduction, models like ours in which inversion frequencies are influenced by balancing selection on pleiotropic tradeoffs may help explain multiple aspects of observed *D. melanogaster* inversion frequencies. This class of models inherently accounts for the intermediate maximal frequency of most *D. melanogaster* inversions in natural populations. To the extent that inversions favor male reproduction, the sexually antagonistic model we propose here could also help account for the predominantly autosomal locations of common inversions in this species, given that male-specific fitness effects have a weaker influence on the frequencies of X-linked variants since the X chromosome spends only one third of its time in males.

This model could also help explain the observed geographic variability of inversion frequencies, as a change in environment may change the fitness value of either pleiotropic effect of the alleles. If a population experiences a novel harsh environment with low survivability then the population density may decrease, and the number of encountered competing males may decrease as well, reducing the relative fitness advantage of enhanced male display. If there is a genetic tradeoff between mating display and viability, the equilibrium frequency of an antagonistic inversion that favors the genetic extreme of display should be lower. Additionally, in hostile environments, the survival cost of display-favoring variants could be increased, which could further shift their equilibrium frequencies downward or render them not worth carrying at any frequency. These predictions could contribute to the observed negative relationships between some inversion frequencies and latitude or altitude (*Kapun et al., 2016*; *Pool et al., 2017*). As a specific example, it is notable that in a high-altitude Ethiopian environment with year-round cool weather, which hosts some of this species' most pronounced phenotypic evolution (*Bastide et al., 2014*; *Lack et al., 2016b*), inversions are virtually absent – occurring at the lowest frequencies of any analyzed population (*Sprengelmeyer et al., 2020*). Concordantly, geographically marginal populations of other *Drosophila* species are often found to host less inversion polymorphism than more central populations (*Krimbas and Powell, 1992*). Also consistent with tradeoffs involving inversion-associated survival costs, a disadvantage of some *D. melanogaster*

inversions in challenging thermal environments (both cold and hot) was supported by a previous experimental evolution analysis that included *In(2 L)t*, *In(2 R)NS*, and *In(3 R)K* (*Kapun et al., 2014*), although they did find that *In(3 R)C* consistently increased in frequency in hot conditions and *In(3 R) Mo* in cold.

In contrast, our experiments yielded evidence that *In(3 R)K* may have a tradeoff opposite from our prediction – with the inverted arrangement associated with greater survival but lower male reproductive success. In line with our findings, *Said et al., 2018* found that among Zambia strains, *In(3 R)K* was associated with strongly reduced expression of *desat2*, involved in pheromone production. Whereas, higher *desat2* expression – as reported by *Said et al., 2018* for standard Zambia karyotypes – has been linked to the 'Z-type' males preferred by females of some strains from this region (e.g. *Fang et al., 2002*; but see *Grillet et al., 2012*). Conversely, a recent genome-wide association study found *In(3 R)K* heterokaryotypes to be associated with elevated male reproductive success compared to standard homokaryotypes, when males from various North Carolina strains were paired with a Z-type Zimbabwe strain known to exhibit preference for southern African male phenotypes (*Yamamoto et al., 2024*), underscoring that inversion-related tradeoffs may depend on genetic background and/ or experimental context.

Given that our model relies on the ability of either an inverted or a standard karyotype to place males carrying it among the top reproductive competitors, it may seem surprising that we found two different inversions associated with a tradeoff between survival and male reproduction – in that an individual's karyotype at one inversion could undercut the advantage of their karyotype the other. In our case, it might seem unfavorable for a male carrying In(3 L)Ok (associated with increased male reproductive performance but lower survival) to also carry In(3 R)K (which showed the opposite patterns). Strikingly then, among 197 sequenced haploid embryo genomes from this Zambia population (each reflecting one female gamete from independent isofemale lines; *Lack et al., 2016a*), not a single genome carried both of these inversions (even though *In(3 L)Ok* and *In(3 R)K* were carried individually by 36 and 35 of these genomes respectively), representing a significant lack of this hypothetically disadvantageous doubly-inverted karyotype (p=0.0031, two-tailed Fisher's Exact Test). These inversions are 20.4 cM apart in terms of meiosis in standard karyotype females (*Comeron et al., 2012*), but the contributions of natural selection versus potentially altered meiotic dynamics in individuals heterozygous for two inversions in generating this pattern will require further investigation.

More broadly, we observed a trend toward increased inversion frequencies across our full experimental generation. A general benefit for inversions would mirror previous findings in which inversions found at intermediate frequencies in natural *D. melanogaster* populations (*Inoue, 1979*), as well as inversions from seaweed flies known to be under balancing selection in the wild (*Mérot et al., 2020*), were both inferred to be under directional selection in the lab. However, one laboratory study did report evidence that a *D. melanogaster* inversion not tested here, *In(3 R)P*, had frequency-dependent fitness under crowded conditions (*Nassar et al., 1973*). Furthermore, even an inversion found to be under directional selection in the lab might still be subject to a balanced tradeoff under a more challenging or complex natural environment. Other studies in *D. melanogaster*, including those on focused seasonal evolution, have found evidence for important tradeoffs between reproduction and robustness to environmental challenges (e.g. *Behrman et al., 2015*). Hence, it would clearly be desirable to study the effects of inversions on a wider range of potential tradeoffs. For example, *Said et al., 2018* suggested that inversions *In(2 L)t* and *In(3 R)K* might be maintained by immunity-related tradeoffs, in light of an over-representation of immune-related genes among differentially expressed transcripts between inverted and standard karyotypes.

In addition to a broader array of balanced tradeoffs that could modulate *D. melanogaster* inversion frequencies, drift and other forms of selection may contribute as well. Although we emphasize above the potential interplay between ecological adaptation and balanced tradeoffs, some inversions could be impacted by simpler forms of directional selection. However, we must then invoke secondary explanations to account for their limited maximal frequencies (see Introduction). Other forms of balancing selection may also come into play as well. It is clear that some variation in this species is subject to temporally varying selection, and some inversions have displayed seasonal frequency differences (*Dobzhansky, 1943*; *Kapun et al., 2016*; *Machado et al., 2021*; *Lange et al., 2022*). As with spatial clines, temporal shifts could reflect either simple directional selection or else

environmentally-modulated tradeoffs in a model such as ours. Spatial selective pressures may vary both between populations and within population scales.

As indicated in the Introduction, balancing selection on inversions could also be achieved by linkage to recessive deleterious variants, which may be observed as an antagonistic pleiotropic phenotype despite potentially involving strictly deleterious variants in linkage (*Pei et al., 2023*). In extreme form inversions might be under associative overdominance (*Zhao and Charlesworth, 2016*; *Gilbert et al., 2020*), where two haplotypes are balanced by recessive cost to both homozygotes. Inversions may be particularly susceptible to carrying recessive deleterious alleles if population size and/or inversion frequency are low, since under these circumstances inversion homozygotes are rarely generated and hence the crossover rate among inverted chromosomes is low, plus there are fewer inverted chromosomes to facilitate rare double recombination events with standard chromosomes. However, this dynamic is not expected to establish new inversions easily (*Charlesworth, 2024*) and is expected to decay with increased age and population size of the inversion, as mutation, gene conversion, or double crossover resolve the negative linkage between recessive deleterious alleles. Most common inversions in *D. melanogaster* are both relatively old and have large effective population sizes. Inversions including *In(2 L)t*, *In(2 R)NS*, and *In(3 R)K* have breakpoints estimated to be several times older than the expansion of the species out of southern-central Africa (*Corbett-Detig and Hartl, 2012*; *Sprengelmeyer et al., 2020*). Those three inversions, plus *In(3 L)Ok*, for which inversion age has not been estimated, all maintain frequencies greater than 0.15 within very large populations (*Kapun and Flatt, 2019*; *Sprengelmeyer et al., 2020*), and so it is not clear that the persistence of recessive deleterious or lethal alleles in the inverted regions is expected to be high enough that inversions would be maintained by a shared recessive lethal variant preventing fixation. However, sampling of natural populations has demonstrated a somewhat higher frequency of recessive lethal variants in inverted haplotypes, but these are largely unique and heterokaryotypic individuals remain robust, giving low costs to a heavily outbred population (*Mukai and Yamaguchi, 1974*). Many wild-derived inbred strains of *D. melanogaster* are homokaryotypic for inversions (*Lack et al., 2016a*). Furthermore, as noted in the Results, an observed lack of inversion homozygotes could reflect strong sexual antagonism as opposed to recessive deleterious variation. Nevertheless, further simulation and experimental testing regarding the potential influence of associative overdominance and other processes on inversion frequencies in *D. melanogaster* and other species would be desirable.

## Generality of balanced pleiotropic inversions

We have proposed that some polymorphic inversions in *D. melanogaster*, and in other species with analogous mating dynamics, may be maintained in part through balancing selection acting on a frequency-dependent tradeoff. While we explored the tradeoff between male reproduction and viability, we emphasize that any antagonistic pleiotropy involving at least one frequency-dependent trait has the potential to generate similar inversion polymorphism. Such antagonistic pleiotropy may be common in evolution, and might be useful in explaining the many examples of inversions associated with distinct phenotypes.

For the specific model explored here, of antagonistic pleiotropy between survival and display in a mate choice system, the prevalence of the dynamic across species and populations is unclear. It seems plausible that a similar model of antagonistic pleiotropy and balancing selection under mate choice underlies other systems, such as the balanced inversion system in the European ruff *Calidris pugnax* (formerly *Philomachus pugnax*), where three inverted haplotypes on an autosome determine aggressive, cooperative, and female-mimic male lekking morphs, albeit with a dominance relationship and homozygous inviability for an inverted haplotype (*Küpper et al., 2016*). A recent study on the female-mimic Faeder morph suggests that it may be maintained in part by a balance between its effects lowering female reproductive output while increasing male mating success, and the system is likely more complex as a whole due to the three interacting morphs (*Giraldo-Deck et al., 2022*). More generally, a number of factors contribute to the potential prevalence of this dynamic. Mate competition and choice during reproduction is rather common among animals, and this may mean sexually antagonistic inversions are plausibly also common. The model depends on the distribution of fitness effects of new mutations and the availability of sexually antagonistic variation, but such variation does seem prevalent (*Zajitschek and Connallon, 2018*; *Ruzicka et al., 2019*). More likely to limit the prevalence of such inversions are the meiotic costs experienced by inversion heterokaryotypes

in some systems (*White, 1973*), whether sexually antagonistic variants are clustered in a region that would allow inversions to increase linkage, and whether other mechanisms like sex-biased expression (*Ingleby et al., 2015*), gene duplication (*Connallon and Clark, 2011*), and linkage to the sex chromosome or sex-determining locus would resolve the antagonism more frequently (though see following subsection on sex chromosome evolution).

Perhaps more importantly, there are a number of well-studied cases of inversion polymorphism that might potentially follow this dynamic of balancing selection between haplotypes simultaneously demonstrating antagonistic pleiotropy and trait-specific frequency dependence, but which do not involve sexual antagonism. An inversion polymorphism in the seaweed fly *Coelopa frigida* has been found to trade off between larval survival and adult reproduction (*Mérot et al., 2020*). *Papilio* butterfly populations harbor well-studied mimicry polymorphisms that segregate between inverted segments and may often experience frequency-dependent selection (*Clarke and Sheppard, 1960*; *Kunte et al., 2014*; *Le Poul et al., 2014*). The snail *Cepaea nemoralis* harbors diverse and distinct shell patterning haplotypes associated with inversions that remain balanced within single sampling locations (*Cook, 2013*; *Surmacki et al., 2013*). Several inversions in different ant species segregate a single-queen colony strategy from a cooperative multi-queen strategy within the same range (*Wang et al., 2013*; *Libbrecht and Kronauer, 2014*). Some vertebrates found to harbor antagonistically pleiotropic inversions include the rainbow trout (*Pearse et al., 2019*) and the zebra finch (*Pei et al., 2023*). These balanced polymorphisms are variously hypothesized to involve maintenance by overdominance (*Mérot et al., 2020*), linked deleterious recessives (*Pei et al., 2023*), or reversals of dominance (*Pearse et al., 2019*), but these mechanisms are not mutually exclusive with each other, or with the model presented here.

Essentially, any population with antagonistic pleiotropy involving some form of negative frequency-dependent balancing selection on one of the phenotypes might exhibit the same dynamic. The population would experience similar divergent selection towards alternate phenotypic extremes, with the frequency-dependence maintaining the presence of two genotypes. Importantly, this model involves a different mechanism for the maintenance of antagonistically pleiotropic variation than models based on reversals in dominance, where dominance effects generate a net heterozygote advantage at a locus (*Connallon and Chenoweth, 2019*). The model proposed here integrates well with existing thought on the evolution of divergent selection on balanced systems. For example, *Kopp and Hermisson, 2006* have proposed a specific form of frequency-dependent divergent selection (FDDS), involving competitive exclusion between similar genotypes for a trait otherwise under stabilizing selection, and this model was expected to result in single or few loci of large effect. Their model did not consider pleiotropy or linkage modifiers, but still had comparable dynamics in that selection in the FDDS regime favored small numbers of large effect loci, while the inversion haplotypes in the model presented here simplify the recombination architecture to approximate a single locus. This simplification of the genetic architecture follows the reduction principle in recombination modifier theory (*Feldman, 1972*; *Feldman and Balkau, 1973*; *Feldman et al., 1980*; *Altenberg and Feldman, 1987*; *Altenberg et al., 2017*). When a population experiences selection towards two (or more) divergent optima, under a selection regime that provides a balancing mechanism to maintain both, indirect selection will favor a reduction of the genetic architectures to prevent the generation of intermediate phenotypes. The likelihood of either model likely rests in part upon the inherent polygenicity of the trait under selection, with simpler traits being more likely to resolve by single locus FDDS dynamics than by inversion association (e.g. *Yassin et al., 2016*).

Notably, our simulations tended to result in strong sexual antagonism, such that it leads to a substantial fraction of the population failing to survive to reproductive age. It is unclear how common such extreme antagonism may be in natural populations. If it is uncommon in nature, even in taxa that experience high levels of mate competition as modeled here, then it is unclear what prevents our simulation predictions from being realized. It may be that genetic constraints prevent the occurrence of highly antagonistic genotypes, or that the relationship between survival-altering mutations and fitness is different than we have modeled. Or, populations may steadily evolve to mitigate antagonistic effects that arise and persist through processes such as those we modeled. Alternatively, some highly antagonistic haplotypes may become associated with sex determination, as suggested below.

## Antagonistic autosomal inversions may precede novel sex chromosomes

If balanced, sexually antagonistic, autosomal inversions occur frequently enough, they may also contribute to the evolution of sex chromosomes. Given an established inversion with significant sexual antagonism, the inverted haplotype that favors one sex still experiences an antagonistic cost of transmission to offspring of the opposite sex. Mutations that link the haplotype with the sex it benefits are therefore favored under selection, and this can occur either by fusing to an existing sex chromosome (*Charlesworth and Charlesworth, 1980*), although this only fully resolves the antagonism for linkage to Y and W chromosomes, or by linkage to a new sex determining mutation (*van Doorn and Kirkpatrick, 2007*).

Here, we discovered that strong inversion-linked antagonism could yield a scenario where essentially all highly-successful males were homokaryotypic for a display-favoring karyotype, whereas reproducing females were overwhelmingly heterokaryotypic (*Figure 5*). This situation is similar to the ZW sex chromosome system, suggesting that this particular pleiotropic trade-off may favor resolution by linkage to a female sex determination factor. Further, simulations in which only females experienced antagonistic survival cost usually generated populations with a chromosome essentially lethal in females, thus exclusively transmitted in males similar to a Y chromosome (*Figure 7*). These simulations sometimes had three states that appeared stable at 20 N generations, including Y chromosome-like, W chromosome-like, and shared intermediate arrangements. This three-arrangement stable polymorphism seems unusual for a natural population, but might be interesting to explore in populations with polymorphic sex determination.

There are often deleterious effects of chromosome fusion or novel sex determination (*Pennell et al., 2015*; *Saunders et al., 2019*), which may represent a general obstacle to sex chromosome evolution. However, balanced sexually antagonistic inversions may concentrate enough antagonism that the built-in benefit of resolving it may sometimes overcome the fitness costs of evolving new sex chromosomes (*Blackmon and Brandvain, 2017*). Hence, antagonistic inversions might be more likely than other loci to be involved in the evolution of new sex chromosomes.

A basic prediction of the above hypothesis is that at least one inversion difference between sex chromosomes should already exist at the time of new sex chromosome formation, and genomic differentiation therefore begins between inversion karyotypes on an autosome even before sex-specific transmission of the region. This scenario contrasts with current models of sex chromosome evolution in which inversion accumulation and genomic differentiation accrue only after sex chromosome formation, due to advantages in suppressing recombination involving sexual antagonistic variants already associated with a sex-determining locus (*Wright et al., 2016*). In this context, we note that the neo-X and neo-Y homologs of Muller element C in *Drosophila miranda*, which became sex-linked only about 1.5 million years ago, already appear to have inversion differences between them (*Wei et al., 2024 – Figure 5A*). Further, a recent study in *Littorina saxatalis* snails has discovered an ecotype-specific sex determination system that may be founded on such tradeoffs, though selection across ecotypes involves discrete geographic boundaries and is only partly comparable (*Hearn et al., 2022*). Still, additional genomic studies of new sex chromosomes are needed to assess the potential role of sexually antagonistic inversions in sex chromosome evolution.

## Summary and future prospects

We have advanced a model in which inversion polymorphism is maintained by the frequency-dependent fitness effects of a pleiotropic tradeoff associated with inversion-linked genetic variation. We have used sexual antagonism associated with increased male reproduction versus survival as a specific motivating case, and we explore the predictions of this model using a novel simulation algorithm. In our simulations, antagonistic variants could be maintained at an equilibrium frequency, and inversions enabled multiple antagonistic variants to persist and form more strongly antagonistic haplotypes. We complemented our conceptual and computational modeling with laboratory experiments designed to detect fitness effects of four inversions common in an ancestral range population of *D. melanogaster*. We indeed found some evidence of tradeoffs between male reproduction and survival for these inversions, with two out of four inversions showing significant evidence for such a tradeoff, although we also observed a strong dependence of male reproductive success on female genetic background.

In light of the persistent mystery of how *D. melanogaster* inversion frequencies are determined, and how readily we detected tradeoffs involving male reproduction and survival, we suggest that further studies investigating the potential fitness tradeoffs of inversions in this model system are strongly warranted. While some such studies have previously been conducted, the applicability of our amplicon sequencing approach to estimate inversion frequencies in large pools of flies may improve the sensitivity of such experiments. Our understanding of the phenotypic impacts of *D. melanogaster* inversions would benefit from investigations of a wider range of fitness-related traits that might be involved in balanced tradeoffs, including female fecundity and immunity. It will also be important to verify the potential frequency dependence and phenotypic consequences of *D. melanogaster* inversions under a wider range of conditions, including challenging environments such as cool temperatures and high population densities. While inherently challenging, a further goal would be to identify loci within an inversion such as *In(3 L)Ok* or *In(3 R)K* that contribute to observed tradeoffs, and in the case of an arrangement favoring male reproduction, to test for the predicted synergistic epistasis (for male reproductive fitness) among linked variants.

Clearly, it will also be important to test the generality of our model by estimating the fitness associations of inversions in a wider range of species. And particularly in the case of species with very recent changes sex chromosomes, it will be of great interest to test whether one or more inversions are already present between these new sex chromosomes, as predicted a model in which sex chromosome evolution involves the resolution of inversion-associated sexual antagonism. Further analytic and simulation work will also be important in clarifying the range of potential scenarios in which balancing selection and antagonistic pleiotropy may lead to the maintenance of polymorphic haplotypes involving inversions (or parts of the genome that rarely recombine), including studies that consider the effects of dominance and that consider tradeoffs other than the one examined here. Collectively, such studies will advance our broader understanding of the roles of balancing selection, pleiotropic tradeoffs, and genome structure in shaping genetic variation and genomic evolution.

## Materials and methods
### Simulations

We developed a new individual-level forward simulator ('SAIsim' for Sexually Antagonistic Inversion simulator) written in python 3.7 for the simulation of inversions and sexually antagonistic alleles at infinite loci, with uniform rates of mutation, crossing over, and gene conversion (per bp, per generation) over a number of independent chromosome arms. The dynamic we were interested in modeling requires direct accounting of pleiotropy between different components of fitness, crossing over and gene conversion within inversions (as well as the possibility of inversion fixation), and directly modeling male display and female choice, and no existing simulator provided this functionality. These simulations were used to establish the potential for the model to occur in a population generally, and not to compare against specific empirical observations. We have used literature estimates of mutation rate and conversion rate of Zambian *D. melanogaster* for these simulations (*Comeron et al., 2012*; *Huang et al., 2016*) but have no information on the joint distribution of effects of mutations on our model of survival and display in *D. melanogaster*. So, we sampled from relatively broad but arbitrary combinations of mutation effects, only assuming that there is likely some degree of antagonistic pleiotropy naturally present in genomic variation.

In SAIsim, a population is instantiated as a python object, and populated with individuals which are also represented by python objects. These individuals may be instantiated using genomes specified by the user, or by default carry no genomic variation. In each generation, as diagrammed in *Figure 2*, the simulator calculates a survival probability for each individual as the product of the survival values of each variant they carry, which range [0,1]. Here, multiplicative selection for survival prevents biologically unreasonable negative survival probabilities from occurring. Male and female survival costs can each be included in a sex-specific manner, but only globally in the simulation, not per mutation. For each offspring in the next generation, a mother is selected randomly, weighted by individual survival probabilities, with replacement. To model the encountered males that might become the father of an offspring individual from this female, a set number of males (i.e. the uniformly specified encounter number, here 100 males unless otherwise specified) is randomly sampled from the full population of males, weighted by their survival probabilities. The males each have a genetic display quality given

as the sum of the reproductive values of their alleles, each of which range [0,1]. A noise parameter, normally distributed around zero with a standard deviation of one, is added to each male display value during each encounter. The male with the highest noise-adjusted display quality among the encountered males is chosen to be the father. As offspring are generated, crossover locations are sampled, and resampled in cases where the resultant gamete would be aneuploid. While the simulator can allow recombination in both sexes, all simulations presented only generate crossovers and gene conversion events for female gametes, in accordance with the biology of *D. melanogaster*. Resampling aneuploids removes the fecundity cost that may exist in some taxa, but was chosen to reflect the lack of observed fecundity effects for most paracentric inversions of *D. melanogaster*. Gene conversion is modeled as a constant probability per heterozygous variant, with each event independently affecting just one locus, rather than explicitly modeling a per-bp conversion rate and tract length (given that gene conversion tracts are generally short and each would typically contain only a single modeled variant in our simulations). The dynamics of gene conversion are expected to have separate effects on the assembly and the maintenance of arrangements. The assembly of an arrangement is positively dependent on the population-scaled rate of recombination of alleles as new polymorphisms start at low frequency and often not on their optimal haplotype. However, the indirect fitness effect of the inversion is dependent on its effect on recombinant offspring, and therefore negatively depends on the per-individual rate of conversion. Finally, inversions and new mutations are potentially added to the resultant gamete, with inversions resampled when overlapping the coordinates of an inversion already present on the same individual chromosome. SAIsim also allows the specification of mutation rates and the distribution of effect sizes of new mutations.

The simulations presented in *Figures 5–7* rescale parameter values by a factor of 2000 to emulate an estimated ancestral $N_e$ of roughly $2 \times 10^6$ (*Sprengelmeyer et al., 2020*) while using a smaller simulated population of 1000 individuals. The low crossover rate simulations model a 0.0218 cM region and the high crossover rate simulations model 2.18 cM, which scale to 43.6 cM and 43.6 M female meiotic maps in the simulated populations. Based on an average autosomal recombination rate in females of $2.18 \times 10^{-8}$ for *D. melanogaster* (*Comeron et al., 2012*), these represent 10 kb and 1 Mb regions. Gene conversion occurs at each heterozygous variant with a probability of 0.1295, based on scaling $\gamma = 1.25 \times 10^{-7}$ events/bp/female meiosis with a conversion track length of 518 bp (*Comeron et al., 2012*) to a per-single-nucleotide-variant estimate of $6.475 \times 10^{-5}$ conversions/bp/female meiosis.

Rates of sexually antagonistic mutations and inversion mutations are empirically undetermined. Here, sexually antagonistic mutations were modeled at a rate corresponding to one per thousand mutations, and so based on an estimate of $5.21 \times 10^{-9}$ mutations/bp/gen on autosomes (*Huang et al., 2016*), and modeling 10 kb and 1 Mb regions, these were scaled to $1.042 \times 10^{-4}$ and $1.042 \times 10^{-2}$ events per gamete respectively. Inversions were approximated as arising once per hundred sexually antagonistic mutations, that is $1.042 \times 10^{-6}$ and $1.042 \times 10^{-4}$ events per gamete. The simulations use mutations with uniformly distributed quality values in the range [0,1]; although distributions skewed toward small effects may be more realistic, our uniform approach avoids the need to invoke a specific but empirically unknown distribution.

SAIsim's object-oriented design allows specification of a population model by a new user familiar with the basics of python scripting. A population object can be parameterized and populated in a few lines of python script, and then parameters can be changed between periods of simulation, allowing bottlenecks and demographic changes as well as changes in mutational parameters between generations. Replicate SAIsim simulations were performed via the UW-Madison Center For High Throughput Computing (CHTC), which manages cycle servers using the HTCondor distributed computing project (*Thain et al., 2005*; *Erlandson and Theisen, 2018*).

## Experimental populations and husbandry

To act as a high-inversion source population for the potential fathers in our experiment, we combined 15 males and 15 females from each of 10 inbred strains derived from wild flies collected in Zambia (*Supplementary file 1a*), and allowed the population to breed for two generations. These 10 strains were chosen to collectively carry the following common inversions: *In(2 L)t*, *In(2 R)NS*, *In(3 L)Ok*, *In(3 R) K*. This high-inversion population may not and need not have been at an equilibrium frequency for any particular inversion; instead it was simply designed to transmit each inversion to offspring at meaningful frequency. We collected males from this second generation to use as the paternal pool in

each of our independent maternal line experiments, and these males were expected to carry a wide range of inversion genotypes. Each batch of high-inversion males was crossed to an equal number of virgin females from a single inbred Zambian strain (*Figure 8*; *Supplementary file 1b*). Four such experiments were performed, each using one inversion-free maternal strain, in clear plexiglass cages (19.3 by 19.3 by 30.0 cm) with a loose mesh covering on one end for access by hand.

Embryos were collected by placing four 7.9 cM petri dishes of grape agar in a population cage for 2 day, then counting and freezing the collected eggs. To obtain adult offspring, six bottles of standard *Drosophila* medium (see below) were placed in a population cage for 1 day. Egg-laying for embryo and adult collections occurred on alternating days over 2 weeks, after which all surviving parents were counted and frozen. From the first day in which flies eclosed in a bottle, adults were collected at the end of each period of a day to 2 days and were separated by sex. These cohorts were aged by weekly transfer to new bottles. The adult offspring were therefore initially separated by sex and by the time after laying at which they eclosed, as well as the time in which they were laid, with different early- or late-eclosing batches from a given cross later pooled before DNA extraction. In our husbandry conditions, the earliest eclosion was 11 days after laying, but this was rare and a first eclosion at 12 days was more common among the bottles. Most flies took several days longer to develop, and we considered flies collected in the first 6 days of a 10 day collection window to be the early-eclosing pool, in order to split the pool sample sizes closer to evenly. The collected adult flies were then aged by sequentially transferring them between bottles weekly for approximately 2.5 months, and then the adult offspring were sexed, counted, and collected. A 10-week aging window was chosen based on the time taken for a control set of four bottles of flies from the paternal outbred population to reach 80% mortality, to reflect the survival and longevity stressors of the laboratory bottle environment. All adult and embryo samples were frozen at –80 °C before DNA extraction.

Since no inversions could be inherited through the mothers, inversion frequencies among successful male gametes could be inferred from their pooled offspring. Therefore, the above sampling scheme allowed us to identify inversion frequencies in the set of potential parents, in embryonic offspring, and in older adults, and so to assess potentially selected changes in inversion frequency across male reproduction or between eggs and aged adults (encompassing viability plus longevity). We chose to age the populations to include longevity as long term survival, which both encompasses fitness challenges that generate mortality at earlier time points, and is relevant to the need to survive during periods of unsuitable environments. Although a natural population from North Carolina was estimated to have an average generation time of about 24 days (including developmental time; *Pool, 2015*), in many climates fly populations cycle seasonally between periods of rapid reproduction and survival for months as adults under reproductive diapause (*Saunders et al., 1990*; *Ragland et al., 2019*). While longevity is a key selective pressure underlying overwintering, the relationship between longevity in permissive lab conditions without diapause and in natural conditions under diapause is unclear (*Schmidt et al., 2005*; *Flatt, 2020*), and our experiment represents just one of many possible ways to examine tradeoffs involving survival.

Unless otherwise noted, all flies in the above experiments were kept in a lab space of 23 °C with around a degree of temperature fluctuation and without an artificially controlled day/night cycle. Light exposure was dependent on the varying use of the space by laboratory workers but amounted to near constant exposure to at least a minimal level of lighting, with some variable light due to indirect lighting from adjacent rooms with exterior windows. All fly food used was prepared in batches of 4.5 L water, 500 mL cornmeal, 500 mL molasses, 200 mL yeast, 54 g agar, 20 mL propionic acid, and 45 mL tegosept 10% (in 95% ethanol). The grape agar was prepared in batches of 500 mL from FlyStuff.com grape agar powder, and wet Red Star yeast was brushed over the agar plates before use to encourage laying.

## Sequencing and data preparation

DNA for each replicate age group was extracted by a phenol-chloroform extraction protocol used previously in preparing DNA for the *Drosophila* Genome Nexus and *Drosophila* Population Genomics Project. We followed protocol 'BPC' used in DPGP phase 1 (presented in *Langley et al., 2012*), except with the addition of proteinase-K during homogenization of adult flies. These protocols are ultimately based on the common quick preparation protocol presented in *Drosophila Protocols* (*Huang et al., 2009*). Inversion genotyping was performed by amplicon sequencing, where amplicons

associated with each inversion were designed to contain at least one single-nucleotide polymorphism in the inversion interior that is fixed for alternate alleles between the inversion karyotypes, while still sharing conserved primer regions across karyotypes to prevent primer bias. Genetic variation in these inversions was assessed using 197 previously sequenced and inversion-called haploid genomes from the same Zambian *D. melanogaster* population studied here, available from the *Drosophila* Genome Nexus (*Lack et al., 2016a*). The amplicon region identification and primer selection was performed with a custom script, which calls Primer3 version 2.4.0 for primer candidate generation (*Koressaar and Remm, 2007*; *Untergasser et al., 2012*; *Kõressaar et al., 2018*). Primers thus selected were ordered from IDT as oligos ligated to Illumina stubby adaptors (*Supplementary file 1c*). Illumina sequencing libraries were then prepared for each sample following the Illumina metagenomic amplicon sequencing protocol *Illumina, 2013*, with 50 ng/µL diluted DNA samples. and sequenced using 150 bp paired end reads on an Illumina NovaSeq to at least a depth of 20,000 reads. Depth was selected to ensure high likelihood of detecting a 0.025 change in inversion frequency from an initial frequency of 0.10, based on simulated resampling of the collected flies, DNA extraction, and sequencing.

Sequence preparation and analysis were performed using custom pipeline scripts. Read trimming, alignment, and QC were performed using bwa version 0.7.17-r1188 (*Li, 2013*), SAMtools version 1.13 (*Li et al., 2009*; *Danecek et al., 2021*), pysam version 0.16.0.1 (https://github.com/pysam-developers/pysam; *Heger et al., 2025*) GATK version 3.4–46 *Van der Auwera and O'Connor, 2020*, and Picard version 1.79 (*Broad Institute, 2019*). Reads were trimmed at the ends to remove segments with base quality less than 20, and read pairs were filtered to retain only those pairs where both reads aligned with mapping quality >20. SNP identity at the sites of interest was called from those remaining reads with base quality >20 at the SNP of interest. Due to the presence of chimeric reads in the amplicon library sequencing, we chose to use a single discriminant SNP to determine each inversion's frequency in the read pool, instead of incorporating haplotype or additional SNP information.

## Statistical analysis of inversion frequencies

Before analyzing the significance of frequency change reversals across three cohorts (which is described below), we assessed the significance of the difference in inversion frequency between just two sampled life stages in the same experiment. For each such comparison, we first calculated a chi-square test statistic (as a convenient metric to use when comparing empirical data with simulated experimental replicates, rather than as a formal test) from the two-by-two table of inverted and non-inverted allele counts at two life stages. Allele counts were estimated by multiplying the inversion-specific SNP allele frequency in the sequencing reads by the allelic sample size based on the number of flies present in the sample. We then generated an approximate distribution of the chi-square statistic when selection is absent (our null hypothesis), using a resampling approach (*Figure 8*), as described in the following paragraph. Comparing the observed statistic to our modeled neutral distribution of test-statistic values allowed us to calculate a p-value for this inversion's frequency change between these two life stages.

To generate the neutral distribution of the chi-square statistic, we took the observed paternal inversion frequencies and then sampled from them to obtain new paternal and offspring read counts to represent a single neutral data set as follows. First, we resampled the collected allelic counts (given in *Supplementary file 1p-q*) to account for sampling variance between replicates. For the paternal samples, we needed to account for pre-sequencing mortality as a mean of 9.83% of males died before collection (a range of 5.11 to 15.77%, *Supplementary file 1b*). So, in generating a new paternal read set, we first modeled down-sampling of the paternal allele sample from the initial census allelic count to the collected allelic count as a hypergeometric sampling. For all other cohorts we used a binomial resampling from the collected allelic count. Second, we used a binomial sampling step to account for the experimental variance introduced by the extraction and library preparation. The extraction and library preparation appeared to introduce variation independent of allele sample size, possibly due to variation introduced during PCR or sample homogenization. This inference was based on analysis of data from control library preparations (*Supplementary file 1r-s*) and from sequencing done in parallel with our experiment from sample fly pools with known inversion frequencies. These control samples included an average of six technical replicates (separate library prep and sequencing replicates from the same flies) from four different expected frequencies for *In(2 L)t*, as well as one expected frequency for each of *In(2 R)NS* and *In(3 R)K*. Due to genotyping and husbandry difficulties we did not

have a control set for *In(3 L)Ok*. Based on the inversion frequency differences observed among these technical replicates, we estimated that taking a binomial sampling of size 316 from the sample allele frequency reflected a maximum likelihood model for the sample-size-independent variation introduced by the library prep and sequencing process. Third, from the preliminary resampled paternal inversion frequency, we then took a binomial sample of size equal to the read count to account for the random sample of sequencing reads, thus yielding resampled paternal read counts (given in *Supplementary file 1p-q*) that accounted for the number of sampled and non-sampled individuals, experimental variance, and depth of coverage. Fourth, to generate resampled offspring read sets, we instead used a binomial sampling of half the offspring alleles from the fully resampled paternal allele frequency for each separate offspring cohort, since the homozygous mothers all contributed non-inverted alleles. Finally, each offspring allele sample was resampled using the same size 316 binomial sampling to account for the variation introduced during this library's preparation, and binomial sampled again with the read count size to account for sampling reads from the library, thus yielding resampled offspring read counts that accounted for the same three factors as for paternal counts.

For a pairwise comparison of inversion frequency between two samples, we calculated the same allele count chi-square statistic as described above for the empirical data for each resampled set of allele counts, generating the full distribution of neutral chi-square test statistics from which we could identify extreme value cutoffs. We then obtained a p-value from the proportion of resampled replicates in which the chi-square statistic was greater than or equal to the empirical value. If the test was directional, we took the proportion of resampled replicates with both a chi-square statistic greater than or equal to the empirical value and a frequency difference in the expected direction.

In order to identify a frequency change reversal between life history stages that could reflect a fitness tradeoff, we tested for the joint significance of two opposing frequency changes using three cohorts: specifically a frequency change in one direction between paternal and embryo samples, followed by a frequency change in the opposite direction between embryo and adult offspring samples. These changes both depend on the shared embryonic cohort in the test, and so are not independent. Therefore, we evaluated the two inversion frequency changes jointly from a set of resampled neutral data sets in which the three cohorts arose from the same resampling process. Here we considered how likely it was that a sample of the three cohorts from the associated estimated neutral model had chi-square values for both the comparison of the fathers to embryos and the embryos to adult offspring that were each equivalent or more extreme in the test direction than the observed data. For example, when testing the significance of an observed paternal-embryo increase and then embryo-adult decrease in frequency, our p-value would be defined by the proportion of resampled replicates in which the frequency changes were in the tested directions of increase then decrease, and their chi-square values were both more extreme than the empirical values. The potential for empirical frequency change reversals to occur in either direction was accounted for at the multiple test correction step as indicated below.

To assess the significance of inversion frequency changes considered across all four maternal line crosses at once, we used Fisher's combined probability test. We applied this to both particular three-cohort frequency change reversals and to the set of two-cohort frequency changes. The inversion trajectory is independent between maternal crosses, but the differences between each cross in maternal line and initial paternal genetic variation make the replicates difficult to compare in a more structured way, so we felt this was an appropriate test for combined significance. In assessing our tradeoff hypothesis, we applied this test only for inversions where it was well-motivated; that is where the averaged frequency change across maternal crosses was in opposite directions between paternal/embryo and embryo/adult comparisons.

To account for multiple hypothesis testing in our cohort comparisons, we used a Benjamini-Yekutieli false discovery rate correction for significance. We applied this correction across the set of Fisher's method combined p-values we generated to assess the relevant frequency change reversal hypotheses. So, this corrected for the multiple hypotheses of two potential tradeoff directions and the four inversions studied, from p-values that already combined evidence from different maternal line crosses. Benjamini-Yekutieli was chosen due to its robustness to the unclear dependence relationships between some of the tests involved. For example, multiple tests of frequency changes from a single cohort (*e.g.* an embryo sample compared against both paternal and aged adult samples) are likely to be positively correlated.

## Acknowledgements

Particular thanks to Shimeng Gao, who assisted with *Drosophila* husbandry and collection of adult flies during the within-generation experiment. We also thank members of the Pool lab and three anonymous reviewers for helpful comments on earlier versions of this manuscript. This research was supported by National Science Foundation Graduate Research Fellowship (DGE-1747503 to CSM), and by the National Institutes of Health (grants R35 GM136306 to JEP, T32 GM007133, and T32 HG002760). This research was performed using the compute resources and assistance of the UW-Madison Center for High Throughput Computing (CHTC) in the Department of Computer Sciences. The CHTC is supported by UW-Madison, the Advanced Computing Initiative, the Wisconsin Alumni Research Foundation, the Wisconsin Institutes for Discovery, and the National Science Foundation, and is an active member of the Open Science Grid, which is supported by the National Science Foundation and the U.S. Department of Energy's Office of Science.

## Additional information

### Funding

| Funder | Grant reference number | Author |
|---|---|---|
| National Institute of General Medical Sciences | R35 GM136306 | John E Pool |
| National Science Foundation Graduate Research Fellowship Program | DGE-1747503 | Christopher S McAllester |
| National Institute of General Medical Sciences | T32 GM007133 | Christopher S McAllester |
| National Human Genome Research Institute | T32 HG002760 | Christopher S McAllester |

The funders had no role in study design, data collection and interpretation, or the decision to submit the work for publication.

### Author contributions

Christopher S McAllester, Conceptualization, Software, Investigation, Methodology, Writing - original draft, Writing – review and editing; John E Pool, Conceptualization, Supervision, Funding acquisition, Methodology, Writing – review and editing

### Author ORCIDs

Christopher S McAllester ⓘ https://orcid.org/0000-0002-4546-1009
John E Pool ⓘ https://orcid.org/0000-0003-2968-9545

Reviewer #1 (Public review): https://doi.org/10.7554/eLife.93338.4.sa1
Reviewer #2 (Public review): https://doi.org/10.7554/eLife.93338.4.sa2
Reviewer #3 (Public review): https://doi.org/10.7554/eLife.93338.4.sa3
Author response https://doi.org/10.7554/eLife.93338.4.sa4

## Additional files

### Supplementary files

Supplementary file 1. Tables containing information for experimental fly counts, primers, amplicons, inversion frequency calls, selection coefficient estimates, and p-values across the comparisons presented. (a) Counts of *D. melanogaster* collected from different inbred Zambian lines to generate the F2 paternal pool. (b) Counts of *D. melanogaster* collected or used at the different stages of the experiments. (c) Primers and amplicon sequences used in assessing inversion frequency from genomic DNA in the *D. melanogaster* lab experiments. Coordinates follow *D. melanogaster*

reference genome release 5. (d) A table of fixed differences between inversions identified from the haploid Zambia genomes from the *Drosophila* Genome Nexus, used for inversion frequency calling. Coordinates follow *D. melanogaster* reference genome release 5. (e) A table of p-values calculated for non-line-combined comparisons between each pair of male vs female cohorts. (f) A table of p-values calculated for non-line-combined comparisons between each pair of early vs late eclosing cohorts. (g) A table of p-values calculated for non-line-combined comparisons between each pair of paternal vs embryo cohorts. (h) A table of p-values calculated for non-line-combined comparisons between each pair of embryo vs aged offspring cohorts.(i) A table of p-values calculated for non-line-combined tests of an increase then decrease across paternal to embryo to aged offspring sets.(j) A table of p-values calculated for non-line-combined tests of a decrease then increase across paternal to embryo to aged offspring sets.(k) A table of p-values for increase-then-decrease tests of paternal-embryo-aged offspring which have been combined across experiments of different maternal inbred line by using fishers combined p-value across maternal lines, then multiple test corrected across tested inversions and directions.(l) A table of p-values for decrease-then-increase tests of paternal-embryo-aged offspring which have been combined across experiments of different maternal inbred line by using fishers combined p-value across maternal lines, then multiple test corrected across tested inversions and directions.(m) A table of p-values for male vs female comparisons combined across experiments of different maternal inbred line by using fishers combined p-value across maternal lines, then multiple test corrected across tested inversions.(n) A table of p-values for early vs late eclosing comparisons combined across experiments of different maternal inbred line by using fishers combined p-value across maternal lines, then multiple test corrected across tested inversions.(o) A table of selection estimates generated by modeling the experimental generation as a Wright-Fisher population (p) Data on the libraries generated from experimental cohorts to estimate the inversion frequencies. Library names represent the maternal line, the cohort, the inversion chromosome arm,.the library prep, and the sequencing run. (q) Data taken by summing the read counts from duplicate libraries generated from experimental cohorts to estimate the inversion frequencies. Pool names represent the maternal line, the cohort, the inversion chromosome arm. (r) Data on the replicate libraries generated from the DNA extractions of experimental cohorts used to estimate the sample-size independent variation introduced by the library preparation and sequencing. Library names represent the source sample or line.the fly pool. the inversion.the extraction and library prep replicate. (s) Data on further replicate libraries including multiple DNA extractions of different inbred fly pools with known inversion frequency, used to estimate bias and the sample-size independent variation introduced by the library preparation and sequencing. Library names represent the source sample or line.the fly pool.the inversion.the extraction and library prep replicate.

MDAR checklist

## Data availability

Sequencing reads have been uploaded to the NIH Sequence Read Archive under BioProject ID PRJNA1213821. The simulation program can be found at https://github.com/csmcal/SAIsim (copy archived at *McAllester, 2017*) and all other analysis scripts can be found at https://github.com/csmcal/dmel_inv_tradeoff (copy archived at *McAllester, 2025*).

The following dataset was generated:

| Author(s) | Year | Dataset title | Dataset URL | Database and Identifier |
|---|---|---|---|---|
| Mcallester CS, Pool JE | 2025 | *D. melanogaster* inversion life-history trade-offs and sexual antagonism | https://www.ncbi.nlm.nih.gov/bioproject/PRJNA1213821 | NCBI BioProject, PRJNA1213821 |

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
