## [Editor Report · eLife Assessment]

This study proposes a new model that could solve some long-standing puzzles about inversion polymorphisms in *Drosophila melanogaster* by invoking sexually antagonism and negative frequency-dependent selection. While the idea developed here is a **valuable** contribution to the field, the experiment only addresses one element of the hypothesis, so that the empirical evidence in support of the model remains **incomplete**.

---

## [Referee Report · Reviewer #1 (Public review)]

The hypothesis is based on the idea that inversions capture genetic variants that have antagonistic effects on male sexual success (via some display traits) and survival of females (or both sexes) until reproduction. Furthermore, a sufficiently skewed distribution of male sexual success will tend to generate synergistic epistasis for male fitness even if the individual loci contribute to sexually selected traits in an additive way. This should favor inversions that keep these male-beneficial alleles at different loci together at a cis-LD. A series of simulations are presented and show that the scenario works at least under some conditions. While a polymorphism at a single locus with large antagonistic effects can be maintained for a certain range of parameters, a second such variant with somewhat smaller effects tends to be lost unless closely linked. It becomes much more likely for genomically distant variants that add to the antagonism to spread if they get trapped in an inversion; the model predicts this should drive accumulation of sexually antagonistic variants on the inversion versus standard haplotype, leading to the evolution of haplotypes with very strong cumulative antagonistic pleiotropic effects. This idea has some analogies with one of predominant hypotheses for the evolution of sex chromosomes, and the authors discuss these similarities. The model is quite specific, but the basic idea is intuitive and thus should be robust to the details of model assumption. It makes perfect sense in the context of the geographic pattern of inversion frequencies. One prediction of the models (notably that leads to the evolution of nearly homozygously lethal haplotypes) does not seem to reflect the reality of chromosomal inversions in *Drosophila*, as the authors carefully discuss, but it is the case of some other "supergenes", notably in ants. So the theoretical part is a strong novel contribution,

To provide empirical support for this idea, the authors study the dynamics of inversions in population cages over one generation, tracking their frequencies through amplicon sequencing at three time points: (young adults), embryos and very old adult offspring of either sex (>2 months from adult emergence). Out of four inversions included in the experiment, two show patterns consistent with antagonistic effects on male sexual success (competitive paternity) and the survival of offspring, especially females, until an old age, which the authors interpret as consistent with their theory.

As I have argued in my comments on previous versions, the experiment only addresses one of the elements of the theoretical hypothesis, namely antagonistic effects of inversions on male reproductive success and other fitness components, in particular of females. Furthermore, the design of this experiment is not ideal from the viewpoint of the biological hypothesis it is aiming to test. This is in part because, rather than testing for the effects of inversion on male reproductive success versus the key fitness components of survival to maturity and female reproductive output, it looks at the effects on male reproductive success versus survival to a rather old age of 2 months. The relevance of survival until old age to fitness under natural conditions is unclear, as the authors now acknowledge. Furthermore, up to 15% of males that may have contributed to the next generation did not survive until genotyping, and thus the difference between these males' inversion frequency and that in their offspring may be confounded by this potential survival-based sampling bias. The experiment does not test for two other key elements of the proposed theory: the assumption of frequency-dependence of selection on male sexual success, and the prediction of synergistic epistasis for male fitness among genetic variants in the inversion. To be fair, particularly testing for synergistic epistasis would be exceedingly difficult, and the authors have now included a discussion of the above caveats and limitations, making their conclusions more tentative. This is good but of course does not make these limitations of the experiment go away. These limitations mean that the paper is stronger as a theoretical than as an empirical contribution.

---

## [Referee Report · Reviewer #2 (Public review)]

Summary:

In their manuscript the authors address the question whether the inversion polymorphism in *D. melanogaster* can be explained by sexually antagonistic selection. They designed a new simulation tool to perform computer simulations, which confirmed their hypothesis. They also show a tradeoff between male reproduction and survival. Furthermore, some inversions display sex-specific survival.

Strengths:

It is an interesting idea on how chromosomal inversions may be maintained

Weaknesses:

The authors motivate their study by the observation that inversions are maintained in *D. melanogaster* and because inversions are more frequent closer to the equator, the authors conclude that it is unlikely that the inversion contributes to adaptation in more stressful environments. Rather the inversion seems to be more common in habitats that are closer to the native environment of ancestral *Drosophila* populations.

While I do agree with the authors that this observation is interesting, I do not think that it rules out a role in local adaptation. After all, the inversion is common in Africa, so it is perfectly conceivable that the non-inverted chromosome may have acquired a mutation contributing to the novel environment.

Based on their hypothesis, the authors propose an alternative strategy, which could maintain the inversion in a population. They perform some computer simulations, which are in line with the predicted behavior. Finally, the authors perform experiments and interpret the results as empirical evidence for their hypothesis. While the reviewer is not fully convinced about the empirical support, the key problem is that the proposed model does not explain the patterns of clinal variation observed for inversions in *D. melanogaster*. According to the proposed model, the inversions should have a similar frequency along latitudinal clines. So in essence, the authors develop a complicated theory because they felt that the current models do not explain the patterns of clinal variation, but this model also fails to explain the pattern of clinal variation.

---

## [Referee Report · Reviewer #3 (Public review)]

Summary:

In this study, McAllester and Pool develop a new model to explain the maintenance of balanced inversion polymorphism, based on (sexually) antagonistic alleles and a trade-off between male reproduction and survival (in females or both sexes). Simulations of this model support the plausibility of this mechanism. In addition, the authors use experiments on four naturally occurring inversion polymorphisms in *D. melanogaster* and find tentative evidence for one aspect of their theoretical model, namely the existence of the above-mentioned trade-off in two out of the four inversions.

Strengths:

(1) The study develops and analyzes a new (*Drosophila melanogaster*-inspired) model for the maintenance of balanced inversion polymorphism, combining elements of (sexually) antagonistically (pleiotropic) alleles, negative frequency-dependent selection and synergistic epistasis. Simulations of the model suggest that the hypothesized mechanism might be plausible.

(2) The above-mentioned model assumes, as a specific example, a trade-off between male reproductive display and survival; in the second part of their study, the authors perform laboratory experiments on four common *D. melanogaster* inversions to study whether these polymorphisms may be subject to such a trade-off. The authors observe that two of the four inversions show suggestive evidence that is consistent with a trade-off between male reproduction and survival.

Open issues:

(1) A gap in the current modeling is that, while a diploid situation is being studied, the model does not investigate the effects of varying degrees of dominance. It would thus be important and interesting, as the authors mention, to fill this gap in future work,

(2) It will also be important to further explore and corroborate the potential importance and generality of trade-offs between different fitness components in maintaining inversion polymorphisms in future work.

---

## [Author Response]

The following is the authors’ response to the current reviews.

**Public Reviews:**

**Reviewer #1 (Public review):**
The hypothesis is based on the idea that inversions capture genetic variants that have antagonistic effects on male sexual success (via some display traits) and survival of females (or both sexes) until reproduction. Furthermore, a sufficiently skewed distribution of male sexual success will tend to generate synergistic epistasis for male fitness even if the individual loci contribute to sexually selected traits in an additive way. This should favor inversions that keep these male-beneficial alleles at different loci together at a cis-LD. A series of simulations are presented and show that the scenario works at least under some conditions. While a polymorphism at a single locus with large antagonistic effects can be maintained for a certain range of parameters, a second such variant with somewhat smaller effects tends to be lost unless closely linked. It becomes much more likely for genomically distant variants that add to the antagonism to spread if they get trapped in an inversion; the model predicts this should drive accumulation of sexually antagonistic variants on the inversion versus standard haplotype, leading to the evolution of haplotypes with very strong cumulative antagonistic pleiotropic effects. This idea has some analogies with one of predominant hypotheses for the evolution of sex chromosomes, and the authors discuss these similarities. The model is quite specific, but the basic idea is intuitive and thus should be robust to the details of model assumption. It makes perfect sense in the context of the geographic pattern of inversion frequencies. One prediction of the models (notably that leads to the evolution of nearly homozygously lethal haplotypes) does not seem to reflect the reality of chromosomal inversions in *Drosophila*, as the authors carefully discuss, but it is the case of some other "supergenes", notably in ants. So the theoretical part is a strong novel contribution.

We appreciate the detailed and accurate summary of our main theoretic results.

To provide empirical support for this idea, the authors study the dynamics of inversions in population cages over one generation, tracking their frequencies through amplicon sequencing at three time points: (young adults), embryos and very old adult offspring of either sex (>2 months from adult emergence). Out of four inversions included in the experiment, two show patterns consistent with antagonistic effects on male sexual success (competitive paternity) and the survival of offspring, especially females, until an old age, which the authors interpret as consistent with their theory.As I have argued in my comments on previous versions, the experiment only addresses one of the elements of the theoretical hypothesis, namely antagonistic effects of inversions on male reproductive success and other fitness components, in particular of females. Furthermore, the design of this experiment is not ideal from the viewpoint of the biological hypothesis it is aiming to test. This is in part because, rather than testing for the effects of inversion on male reproductive success versus the key fitness components of survival to maturity and female reproductive output, it looks at the effects on male reproductive success versus survival to a rather old age of 2 months. The relevance of survival until old age to fitness under natural conditions is unclear, as the authors now acknowledge. Furthermore, up to 15% of males that may have contributed to the next generation did not survive until genotyping, and thus the difference between these males' inversion frequency and that in their offspring may be confounded by this potential survival-based sampling bias. The experiment does not test for two other key elements of the proposed theory: the assumption of frequency-dependence of selection on male sexual success, and the prediction of synergistic epistasis for male fitness among genetic variants in the inversion. To be fair, particularly testing for synergistic epistasis would be exceedingly difficult, and the authors have now included a discussion of the above caveats and limitations, making their conclusions more tentative. This is good but of course does not make these limitations of the experiment go away. These limitations mean that the paper is stronger as a theoretical than as an empirical contribution.

We discuss the choice to focus on exploring the potential antagonistic effects of the inversion karyotype on male reproductive success and survival in our general response above. Primarily, this prediction seemed to be the most specific to the proposed model as compared to other alternate models. Still, further studies are clearly needed to elucidate the potential frequency dependence and genetic architecture of the inversions.

Regarding the choice of age at collection, it is unknown to what degree our selected collection age of 10 weeks correlates with survival in the wild, but we feel confident that there will be some positive correlation.

We now further clarify that across our experiments, a minimum of 5% and a mean of 9% of the males used in the parental generation died before collection. These proportions do not appear sufficient to explain the differences between paternal and embryo inversion frequencies shown in Figure 9.

**Reviewer #2 (Public review):**
Summary:In their manuscript the authors address the question whether the inversion polymorphism in *D. melanogaster* can be explained by sexually antagonistic selection. They designed a new simulation tool to perform computer simulations, which confirmed their hypothesis. They also show a tradeoff between male reproduction and survival. Furthermore, some inversions display sex-specific survival.Strengths:It is an interesting idea on how chromosomal inversions may be maintainedWeaknesses:The authors motivate their study by the observation that inversions are maintained in *D. melanogaster* and because inversions are more frequent closer to the equator, the authors conclude that it is unlikely that the inversion contributes to adaptation in more stressful environments. Rather the inversion seems to be more common in habitats that are closer to the native environment of ancestral Drosophila populations.While I do agree with the authors that this observation is interesting, I do not think that it rules out a role in local adaptation. After all, the inversion is common in Africa, so it is perfectly conceivable that the non-inverted chromosome may have acquired a mutation contributing to the novel environment.Based on their hypothesis, the authors propose an alternative strategy, which could maintain the inversion in a population. They perform some computer simulations, which are in line with the predicted behavior. Finally, the authors perform experiments and interpret the results as empirical evidence for their hypothesis. While the reviewer is not fully convinced about the empirical support, the key problem is that the proposed model does not explain the patterns of clinal variation observed for inversions in *D. melanogaster*. According to the proposed model, the inversions should have a similar frequency along latitudinal clines. So in essence, the authors develop a complicated theory because they felt that the current models do not explain the patterns of clinal variation, but this model also fails to explain the pattern of clinal variation.

To the contrary – in the Discussion paragraph beginning on Line 671, we explain why we would predict that a tradeoff between survival and reproduction should lead to clinal inversion frequencies. We suggest that a karyotype associated with a survival penalty should be increasingly disadvantageous in more challenging environments (such as high altitudes and latitudes for this species). Furthermore, an advantage in male reproductive competition conferred by that same haplotype may be reduced by the lower population densities that we would expect in more challenging environments (meaning that each female should encounter fewer males). Individually or jointly, these two factors predict that the equilibrium frequency of a balanced inversion frequency polymorphism should depend on a local population’s environmental harshness and population density, with the ensuing prediction that inversion frequency should correlate with certain environmental variables.

**Reviewer #3 (Public review):**
Summary:In this study, McAllester and Pool develop a new model to explain the maintenance of balanced inversion polymorphism, based on (sexually) antagonistic alleles and a trade-off between male reproduction and survival (in females or both sexes). Simulations of this model support the plausibility of this mechanism. In addition, the authors use experiments on four naturally occurring inversion polymorphisms in *D. melanogaster* and find tentative evidence for one aspect of their theoretical model, namely the existence of the above-mentioned trade-off in two out of the four inversions.Strengths:(1) The study develops and analyzes a new (*Drosophila melanogaster*-inspired) model for the maintenance of balanced inversion polymorphism, combining elements of (sexually) antagonistically (pleiotropic) alleles, negative frequency-dependent selection and synergistic epistasis. Simulations of the model suggest that the hypothesized mechanism might be plausible.(2) The above-mentioned model assumes, as a specific example, a trade-off between male reproductive display and survival; in the second part of their study, the authors perform laboratory experiments on four common *D. melanogaster* inversions to study whether these polymorphisms may be subject to such a trade-off. The authors observe that two of the four inversions show suggestive evidence that is consistent with a trade-off between male reproduction and survival.Open issues:(1) A gap in the current modeling is that, while a diploid situation is being studied, the model does not investigate the effects of varying degrees of dominance. It would thus be important and interesting, as the authors mention, to fill this gap in future work.(2) It will *also be important to further explore and corroborate the potential importance and generality of trade-offs between different fitness components in maintaining inversion polymorphisms in future work.*

We appreciate the work put in to evaluating, improving, and summarizing our study. We agree that further work studying the effects of dominance and of the fitness components of the inversions is important.

**Recommendations for the authors:**

**Reviewer #1 (Recommendations for the authors):**
l. 354 : I don't understand what the authors mean by "an antagonistic and non-antagonistic allele". If there is a antagonistic polymorphism at a locus, then both alleles have antagonistic effects; i.e., allele B increases trait 1 and reduced trait 2 relative to allele A and vice versa.

Edited, agreed that the terminology used here was sub-optimal.

**Reviewer #2 (Recommendations for the authors):**
The motivation for their model is their claim that the clinal inversion frequencies are not compatible with local adaptation. The reviewer doubts this strong statement. Furthermore, the proposed model also fails to explain the inversion frequencies in natural populations.Hence, rather than building a straw man, it would be better if the authors first show their experiments and then present their model as an explanation for the empirical results. Nevertheless, it is also clear that the empirical data are not very strong and cannot be fully explained by the proposed model.

This claim that we reject any role of local adaptation in clinal variation and selection upon inversion polymorphism does not hold up in a reading of our manuscript. We even suggest that locally varying selective pressures must be playing some role, although that does not imply that local adaptation is the ultimate driver of inversion frequencies. Indeed, we suggest that local adaptation alone is an insufficient explanation for inversion frequency clines in *D. melanogaster*, including because (1) these frequency clines do not approach the alternate fixed genotypes predicted by local directional selection, (2) these derived inversions tend to be more frequent in more ancestral environments (l.113-158).

In our public review response above, and in the Discussion section of our paper, we explain why our model can predict both the clinal frequencies of many *Drosophila* inversions and their intermediate maximal frequencies. Of course, we do not predict that most inversions in this species should follow the specific tradeoff investigated here. In fact, we were surprised to find even two inversions that experimentally supported our predicted tradeoff. Still, it remains possible that other inversions in this species are subject to other balanced tradeoffs not investigated here, which could help explain why they rarely reach high local frequencies.

**Reviewer #3 (Recommendations for the authors):**
My previous comments have been adequately addressed.

The following is the authors’ response to the original reviews.

**Reviewer #1 (Public Review):**
[…]To provide empirical support for this idea, the authors study the dynamics of inversions in population cages over one generation, tracking their frequencies through amplicon sequencing at three time points: (young adults), embryos and very old adult offspring of either sex (>2 months from adult emergence). Out of four inversions included in the experiment, two show patterns consistent with antagonistic effects on male sexual success (competitive paternity) and the survival of offspring, especially females, until an old age, which the authors interpret as consistent with their theory.There are several reasons why the support from these data for the proposed theory is not waterproof.(1) As I have already pointed out in my previous review, survival until 2 months (in fact, it is 10 weeks and so 2.3 months) of age is of little direct relevance to fitness, whether under natural conditions or under typical lab conditions.The authors argue this objection away with two argumentsFirst, citing Pool (2015) they claim that the average generation time (i.e. the average age at which flies reproduce) in nature is 24 days. That paper made an estimate of 14.7 generations per year under the North Carolina climate. As also stated in Pool (2015), the conditions in that locality for Drosophila reproduction and development are not suitable during three months of the year. This yields an average generation length of about 19.5 days during the 9 months during which the flies can reproduce. On the highly nutritional food used in the lab and at the optimal temperature of 25 C, *Drosophila* need about 11-12 days to develop from egg to adult. Even assuming these perfect conditions, the average age (counted from adult eclosion) would be about 8 days. In practice, larval development in nature is likely longer for nutritional and temperature reasons, and thus the genomic data analyzed by Pool imply that the average adult age of reproducing flies in nature would be about 5 days, and not 24 days, and even less 10 weeks. This corresponds neatly to the 2-6 days median life expectancy of Drosophila adults in the field based on capture-recapture (e.g., Rosewell and Shorrocks 1987).Second, the authors also claim that survival over a period of 2 month is highly relevant because flies have to survive long periods where reproduction is not possible. However, to survive the winter flies enter a reproductive diapause, which involves profound physiological changes that indeed allow them to survive for months, remaining mostly inactive, stress resistant and hidden from predators. Flies in the authors' experiment were not diapausing, given that they were given plentiful food and kept warm. It is still possible that survival to the ripe old age of 10 weeks under these conditions still correlates well with surviving diapause under harsh conditions, but if so, the authors should cite relevant data. Even then, I do not think this allows the authors to conclude that longevity is "the main selective pressure" on *Drosophila* (l. 936).

This is overall a thoughtfully presented critique and we have endeavored to improve our discussion of Pool (2015) and to clarify some of the language used about survival elsewhere. While we agree that challenges other than survival to 10 weeks are very relevant to *Drosophila melanogaster*, collection at 10 weeks does encompass some of these other challenges. Egg to adult viability still contributes to the frequencies of the inversions at collection and is not separable from longevity in this data. Collection at longevity was chosen in part to encompass all lifetime fitness challenges that might influence the inversion frequency at collection, albeit still within permissive laboratory conditions. Future experiments exploring specific stressors independently and beyond permissive lab conditions would generate a clearer picture.

In addition to general edits, the specific phrase mentioned at 1. 936 [now line 1003] has been revised from “In many such cases females are in reproductive diapause, and so longevity is the main selective pressure.” to “While longevity is a key selective pressure underlying overwintering, the relationship between longevity in permissive lab conditions without diapause and in natural conditions under diapause is unclear (Schmidt et al. 2005; Flatt 2020), and our experiment represents just one of many possible ways to examine tradeoffs involving survival.”

(2) It appears that the "parental" (in fact, paternal) inversion frequency was estimated by sequencing sires that survived until the end of the two-week mating period. No information is provided on male mortality during the mating period, but substantial mortality is likely given constant courtship and mating opportunities. If so, the difference between the parental and embryo inversion frequency could reflect the differential survival of males until the point of sampling rather than / in addition to sexual selection.

We have further clarified that when referenced as parental frequency, the frequency presented is ½ the paternal frequency as the mothers were homokaryotypic for the standard arrangement. We chose to present both due to considerations in representing the frequency change from paternal to embryo frequencies, where a hypothetical change from 0.20 frequency in fathers to 0.15 frequency in embryos represents a selective benefit (a frequency increase in the population), despite the reality that this is a decrease in allele frequency between paternal and embryo cohorts.

We mentioned a maximum 15% paternal mortality at line 827 [now l.1056], but have now added complete data on the counts of flies in the experiment as a supplemental table (Table S1) and have added or corrected further references to this in the results and methods [lines 555, 638, 975]. It is true that this may influence the observed frequency changes to some degree, and while we adjusted our sampling method to account for the effects of this mortality on statistical power [l.1056ff], we have now edited the manuscript to better highlight potential effects of this phenomenon on the recorded frequency changes.

It is also worth noting that, if mortality among fathers over the mating period is codirectional with mortality among aged offspring, this would bias the results against detecting an opposing antagonistic selective effect of the inversions on paternity share. This is now also mentioned in the manuscript, l.639ff.

(3) Finally, irrespective of the above caveats, the experimental data only address one of the elements of the theoretical hypothesis, namely antagonistic effects of inversions on reproduction and survival, notably that of females. It does not test for two other key elements of the proposed theory: the assumption of frequency-dependence of selection on male sexual success, and the prediction of synergistic epistasis for male fitness among genetic variants in the inversion. To be fair, particularly testing the latter prediction would be exceedingly difficult. Nonetheless, these limitations of the experiment mean that the paper is much stronger theoretical than empirical contribution.

This is a fair criticism of the limitations of our results, and we now summarize such caveats more directly in the discussion summary, lines 876ff.

**Reviewer #2 (Public Review):**
[…]Comments on the latest version:I would like to give an example of the confusing terminology of the authors:"Additionally, fitness conveyed by an allele favoring display quality is also frequency-dependent: since mating success depends on the display qualities of other males, the relative advantage of a display trait will be diminished as more males carry it..."I do not understand the difference to an advantageous allele, as it increases in frequency the frequency increase of this allele decreases, but this has nothing to do with frequency dependent selection. In my opinion, the authors re-define frequency dependent selection, as for frequency dependent selection needs to change with frequency, but from their verbal description this is not clear.

We have edited this text for greater clarity, now line 232ff. We did not seek to redefine frequency dependence, and did mean by “the relative advantage of a display trait will be diminished” that an equivalent s would diminish with frequency. We have now remedied terminological issues introduced in the prior revision with regard to frequency dependent selection.

One example of how challenging the style of the manuscript is comes from their description of the DNA extraction procedure. In principle a straightforward method, but even here the authors provide a convoluted uninformative description of the procedure.

We have edited for clarity the text on lines 1016-1020. Citing a published protocol and mentioning our modifications seems an appropriate trade-off between representing what was done accurately, citing the sources we relied on in doing it, and limiting the volume of information in the main text for such a straightforward and common method.

It is not apparent to the reviewer why the authors have not invested more effort to make their manuscript digestible.

We have invested a great deal of effort in making this manuscript as clear as we are able to. We regret that our writing has not been to this reviewer’s liking. We believe we have been highly responsive to all specific criticisms, including revising all passages cited as unclear. In this round, we have again scrutinized the entire manuscript for any opportunity to clarify it, and we have made further changes throughout. Although our subject matter is conceptually nuanced, we nevertheless remain optimistic that a careful, fresh reading of our revised manuscript would yield a more favorable impression.

**Reviewer #3 (Public Review):**
[…]Weaknesses:A gap in the current modeling is that, while a diploid situation is being studied, the model does not investigate the effects of varying degrees of dominance. It would be important and interesting to fill this gap in future work.

Agreed, and now reinforced at lines 892ff.

Comments on the latest version:Most of the comments which I have made in my public review have been adequately addressed.Some of the writing still seems somewhat verbose and perhaps not yet maximally succinct; some additional line-by-line polishing might still be helpful at this stage in terms of further improving clarity and flow (for the authors to consider and decide).

We have made further changes and some polishing in this draft, and greatly appreciate the guidance provided in improving the draft so far.

**Reviewer #1 (Recommendations For The Authors):**
(1) While the model results are convincing, some of the verbal interpretation is confusing. In particular, the authors state that in their model the allele favoring male display quality shows a negative frequency dependence whereas the alternative allele has a positive frequency dependence. This does not make sense to me in the context of population genetics theory. For a one-locus, two-allele model the change of allele frequency under selection depends on the fitness of the genotypes concerned relative to each other. Thus, at least under no dominance assumed in this model, if the relative fitness of AA decreases with the frequency of allele A, the relative fitness of aa must decrease with the frequency of allele a. I.e., if selection is negatively frequency dependent, then it is so for both alleles.

This phrasing was wrong, and we have edited the relevant section.

(2) I am still not entirely sure that the synergistic epistasis assumed in the verbal model is actually generated in the simulations; this would be easy enough to check by extracting the mating success of males with different genotypes from the simulation output should be reported, e.g., as a figure supplement.

Our new Figure S2, which depicts haplotype frequencies for a set of the simulations presented in Figure 4, should demonstrate a necessary presence of synergistic epistasis. These results further clarify that the weaker allele B is only kept when linked to A. The same fitness classes of genotype are present in the simulations with and without the inversion, so the only mechanical difference is the rate of recombination, and the only way this might change selection on the alleles is if a variant has a different fitness in one haplotype background than another – i.e. epistasis. The maintenance of haplotypes AB and ab to the exclusion of Ab and aB relies on the lesser relative fitness of Ab and aB. And since survival values are multiplicative, this additional contribution must come from the mate success of AB being disproportionately larger than Ab or aB, indicating the emergent synergistic epistasis posited by our model. We have clarified this point in the text at line 363ff.

(3) l. 318ff: What was this set number of males? I could not find this information anywhere. Also, this model of the mating system is commonly referred to as "best of N", so the authors may want to include this label in the description.

We indicate this detail just after the referenced line, now reworded and on l. 338-340 as “For each female’s mating competition, 100 males were sampled, though see Figure S1 for plots with varying encounter number.” Among these edits, “one hundred” has been changed to a numeral for easier skimming, and Figure S1 is now referenced here earlier in the text. Several edits have also been made in the caption of Figures 2 and 3, and in the relevant methods section to clarify the number of encountered males simulated, mention best of N terminology, and clarify how the quality score is used in the mate competition.

(4) The description of the experiment is still confusing. The number of individuals of each sex entered in each mating cage is missing from the Methods (l. 914); although I did finally find it in the Results. These flies were laying over 2 weeks - does this mean that offspring from the entire period were used to obtain the embryo and aged offspring frequencies, or only from a particular egg collection? If the former, does this mean that the offspring obtained from different egg batches were aged separately? Were the offspring aged in cages or bottles, at what density? Given that only those males that survived until the end of the two-week mating period were sequenced, it is important to know what % of the initial number of males these survivors were. A substantial mortality of the parental males could bias the estimate of parental frequencies. How many parental males, embryos and aged offspring were sequenced? Were all individuals of a given cage and stage extracted and sequenced as a single pool or were there multiple pools? The description could also be structured better. For example, the food and grape agar recipes and cage construction are inserted at random points of the description of the crossing design, which does not help.

We have now reorganized and edited these portions of the Methods text. Portions of this comment overlap with edits responding to (2) of the Public Review and below for l. 921 in Details. Offspring from different laying periods were aged in different bottles, further separated by the time at which they eclosed. They were then pooled for DNA extraction and library preparation by sex and a binary early or late eclosion time. This data was present in the “D. mel. Sample Size” column of supplemental tables S6 and S7 (now S7 and S8), but we have added and referenced a new table to specifically collate the sample sizes of different experimental stages, table S1. Now referenced at lines 555, 638, 975, 1057.

(5) The caption of figure 9 and the discussion of its results should be clear and explicit about the fact that "adult offspring" in Fig 9A and "female" and "male" refers to adults surviving to old age whereas "parental" in Fig 9A refers to young adults in their reproductive prime. This has consequences for the interpretation of the difference between "parental" and "adult offspring", as it combines one generation of usual selection as it occurs under the conditions of the lab culture (young adult at generation t -> young adult in generation t+1) with an additional step of selection for longevity. Thus, a marked change in allele frequency does not imply that the "parental" frequency does not represent an equilibrium frequency of the inversions under the lab culture conditions. Furthermore, it would be useful to state explicitly that Figure 9B represents the same results as figure 9A, but with the aged offspring split by sex.

Figure caption edited to provide further clarity on the age of cohorts and presented data, along with the relevant results section (2.3) referencing this figure.

We avoid making any statements about the equilibrium frequencies of inversions under lab conditions, and whether or not any step of our experiment reflects such equilibria, because our investigation does not rely upon or test for such conditions. Instead, our analysis focuses on whether inversions have contrasting effects (as indicated by frequency changes that are incompatible with neutral sampling) between different life history components. Under our model, such frequency reversals might be detectable both at equilibrium balanced inversion frequencies and also at frequencies some distance away from equilibria. We have now clarified this point at l. 970-972.

Details:l. 211: this should be modified as male-only costs are now included.

Edited. “survival likelihood (of either or both sexes).”

l. 343: misplaced period

Edited.

l. 814: "We confirmed model predictions...": This sounds like it refers to an empirical confirmation of a theory prediction, but I think the authors just want to say that their simulations predicted antagonistic variants can be maintained at an intermediate equilibrium frequency. So the wording should be changed to avoid ambiguity.

Edited. Now line 869.

l. 853: How can a genome be "empty"? Do the authors mean an absence of any polymorphism?

Edited to: “In SAIsim, a population is instantiated as a python object, and populated with individuals which are also represented by python objects. These individuals may be instantiated using genomes specified by the user, or by default carry no genomic variation.” Lines 913ff.

l. 853: I do not see this diagramed in Figure 5

Apologies, fixed to Fig. 2

l. 864: is crossing-over in the model limited to female gametogenesis (reflecting the *Drosophila* case) or does it occur in both sexes?

There is a variable in the simulator to make crossover female-specific. All simulations were performed with female-only crossover. Edited for clarity. “While the simulator can allow recombination in both sexes, all simulations presented only generate crossovers and gene conversion events for female gametes, in accordance with the biology of *D. melanogaster*.” Lines 928-929.

l. 906: "F2" is ambiguous; does this mean that the mix of lines was allowed to breed for two generations? Also, in other places in the manuscript these flies appear to be referred to are "parental". So do not use F2.

Edited, F2 language removed and replaced with being allowed to breed for two generations. Now lines 967ff.

l. 910: this is incorrect/imprecise; what can be inferred is the frequency of the inversions in male gametes that contributed to fertilization. This would correspond to the frequency in successful males only if each successful male genotype had the same paternity share.

Edited, now “Since no inversions could be inherited through the mothers, inversion frequencies among successful male gametes could be inferred from their pooled offspring.” Now line 994.

l. 912: "without a controlled day/night cycle" meaning what? Constant light? Constant darkness? Daylight falling through the windows?

Edited to “Unless otherwise noted, all flies were kept in a lab space of 23°C with around a degree of temperature fluctuation and without a controlled day/night cycle. Light exposure was dependent on the varying use of the space by laboratory workers but amounted to near constant exposure to at least a minimal level of lighting, with some variable light due to indirect lighting from adjacent rooms with exterior windows.” Now lines 1007-1010.

l. 921: I cannot parse this sentence. Were the offspring isolated as virgins?

No, the logistics of collecting virgins would have been prohibitive, and it did not seem essential for our experiment. Hopefully the edits to this section are clearer, now lines 978ff.